# DISTILL GOLD FROM MASSIVE ORES: EFFICIENT DATASET DISTILLATION VIA CRITICAL SAMPLES SELECTION

## ABSTRACT

Data-efficient learning has drawn significant attention, especially given the current trend of large multi-modal models, where dataset distillation can be an effective solution. However, the dataset distillation process itself is still very inefficient. In this work, we model the distillation problem with reference to information transport. Observing that severe data redundancy exists in dataset distillation, we argue to put more emphasis on the utility of the training samples. We propose a family of methods to exploit the most valuable samples, which is validated by our comprehensive analysis of the optimal data selection. The new strategy significantly reduces the training cost and extends a variety of existing distillation algorithms to larger and more diversified datasets, *e.g.*, in some cases only **0.04%** training data is sufficient for comparable distillation performance. Moreover, our strategy consistently enhances the performance, which may open up new analyses on the dynamics of distillation and networks. Our method is able to extend the distillation algorithms to much larger-scale datasets and more heterogeneous datasets, *e.g.*, ImageNet-1K and Kinetics-400. *Our code will be made publicly available*.

## 1 INTRODUCTION

Data is crucial for deep learning. Data-*efficient* learning has become critical which enables high performance with less cost, especially in the era of large data and models (Brown et al., 2020; Schuhmann et al., 2021) when the size and complexity of models continue to grow. Techniques such as pruning, quantization, and knowledge distillation have been used to reduce the model size without sacrificing performance. Recently, *dataset distillation* (Wang et al., 2018) has become a promising way towards data-efficient AI, where a small and condensed dataset (*synthetic*) is learned from the whole large dataset (*real*), maintaining the model performance trained on the synthetics.

However, the efficiency of dataset distillation itself poses a major obstacle to its own application. Currently, the distillation process takes substantially more time than training a model on the full dataset, which is some kind of "*penny-wise and pound-foolish*". For example, the training of expert trajectories for MTT (Cazenavette et al., 2022) takes **100×** longer than training a model on the full real dataset. The prohibitive computation and memory requirement make dataset distillation not applicable as the data scale and instance-per-class (IPC) increase, especially on modern large-scale models and datasets whose size can grow to the order of billions (Brown et al., 2020; Radford et al., 2021; Schuhmann et al., 2021). Existing works address efficient distillation by reducing the storage burden of synthetic data, data compression (Kim et al., 2022b; Liu et al., 2022; Qiu et al., 2022; Deng & Russakovsky, 2022), or optimizing the training workflow (Cui et al., 2022), while no adequate works have addressed the efficiency issue of *data utilization*.

In this paper, we seek a novel efficient strategy to exploit massive real data, by first modeling the dataset distillation problem from the information transport point of view. Specifically, the distillation process can be regarded as a *"knowledge" flow* from the real dataset to the synthetic dataset, and finally to the network. Since the synthetic data is much smaller compared to the real data, its capacity is limited, and thus the knowledge flow meets a "**bottleneck**" at the synthetic data. Thus, a natural assumption comes up: we can choose the most valuable real samples (have the most knowledge)

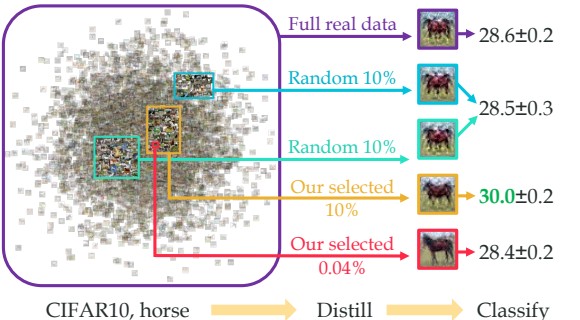 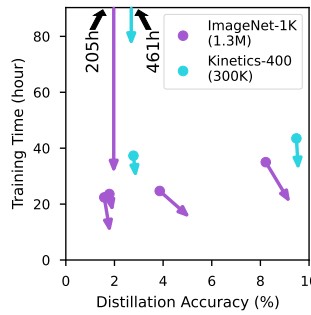

Figure 1: Severe data redundancy in dataset distillation. (1) Left: with our optimal selection policy, only 0.04% real samples are sufficient for distillation and 10% optimal real samples outperform the full real dataset. (2) Right: each arrow indicates an algorithm, pointing from the performance on full data to that with our strategy. Our strategy enables the distillation of large-scale heterogeneous datasets and significantly reduces the training cost while maintaining or enhancing the performance.

and drop the others without blocking the knowledge flow. The slimmest real data subset can then be obtained encompassing enough knowledge for distillation.

Based on this perspective, we observe that real data can be highly *redundant* and insensitive to data reduction in dataset distillation. For instance, with CIFAR10 for DC (Zhao et al., 2020) and single instance-per-class (IPC=1), randomly removing **90%** of the real data does not affect its performance. More importantly, if carefully selected, **even 20 images (0.04% of full dataset) suffice for comparable performance**. This phenomenon widely exists among various datasets, networks, methods, and IPCs, which we will detail in Sec. 3.1. This observation supports the feasibility of *data-efficient distillation* that enables us to deal with massive real datasets. We argue that leveraging data redundancy bears more significance for dataset distillation due to the "knowledge bottleneck".

In light of these observations, we provide a thorough analysis and study on data-efficient distillation. We first define the knowledge content of a set of data samples in terms of its **data utility**, indicating the value or quality of the data. To efficiently and effectively find the optimal real data subset, we propose a family of utility estimation and optimal data selection policies. We compare various criteria to encode the utility of data samples, including data density, gradient, training loss, *etc*, and derive certain criteria that are shown to be especially effective. Our strategy can greatly reduce the training cost and enable larger-scale experiments with shorter training times.

Furthermore, we find that careful data selection can even enhance the distillation sometimes, which indicates that some samples may be "harmful" to the training. With the data utility, we propose a simple but effective baseline tool to exploit the bias toward high-utility samples, to maximize the distillation performance. Our method can achieve stable performance gain without the pre-computation of data utility. With better efficiency, we can handle the whole ImageNet-1K (Deng et al., 2009) and even more heterogeneous data such as videos from Kinetics (Carreira & Zisserman, 2017). Our work is the very first that distills the Kinetics-400 dataset with acceptable efficiency. We provide deeper insight into the internal mechanism of dataset distillation and neural network training.

Overall, our contributions are: 1) we propose the first work on analyzing and modeling the data-efficient distillation to greatly reduce the computation complexity; 2) we propose estimators and methods to exploit the data utility to advance dataset distillation; 3) using the proposed methods, we can efficiently distill very large-scale datasets.

## 2 RELATED WORK

**Dataset Distillation** is a process of compressing a large dataset into a smaller and more representative dataset while maintaining the performance. The existing approaches can be roughly classified into: 1) **Meta-Model Matching** maintains the transferability of the synthetic data, by optimizing the empirical loss on the original dataset of models trained on the synthetic data. Wang et al. (2018) first propose the task of data distillation and use the meta-model matching framework for optimization. Nguyen et al. (2020) exploit kernel ridge regression to facilitate its inner optimization loop, and is

further extended to infinite wide networks (Nguyen et al., 2021). Zhou et al. (2022) separate the optimization of synthetic data/classifier and feature extractor. 2) **Gradient Matching** (Zhao et al., 2020) aligns the gradients of the synthetic and real dataset and is further improved by Zhao & Bilen (2021) to perform the same image augmentations on both the real and synthetic data. 3) **Distribution Matching** (Zhao & Bilen, 2023) matches the feature distributions of the synthetic and real data, which is simple but effective. Wang et al. (2022) design layer-wise feature alignment and a few early exit conditions to promote DM. Zhao et al. (2023) further enhance DM with regularizers and model pool. 4) **Trajectory Matching:** Cazenavette et al. (2022) propose to match the training trajectory of the model parameters, by aligning the future parameters trained on real data and synthetic data. Cui et al. (2022) reduce the memory consumption of MTT and exploit label learning. 5) **Factorization** of synthetic data can reduce the storage burden and share knowledge among instances. Kim et al. (2022b) use a strategy of putting multiple images on one synthetic sample. Deng & Russakovsky (2022) decompose the synthetic data to the linear combination of bases. Liu et al. (2022) use a hallucination network to combine the bases. Lee et al. (2022) maintains a smaller base space to further reduce the storage. 6) **Bayesian Pseudocoreset** is a family of algorithms that learn the synthetic data with Bayesian inference (Manousakas et al., 2020; Kim et al., 2022a; Tiwary et al., 2023).

**Data Selection/Pruning** reduces the training data without significantly affecting performance. Classic data selection often calculates a scalar utility score for each sample based on predefined criteria (Castro et al., 2018; Sener & Savarese, 2017; Toneva et al., 2018) and filters the samples based on scores. Some data pruning methods also consider the interaction between samples. Yang et al. (2022) examines generalization influence to reduce training data, which aims to identify the smallest subset to satisfy the expected generalization ability. In comparison, data distillation (Wang et al., 2018) and data condensation (Zhao et al., 2020) synthesize new and smaller data. The performance of data distillation with the same images per class (IPC) significantly outperforms data pruning.

## 3 PRELIMINARIES

Data redundancy widely exists in various machine learning tasks. After conducting thorough experiments and comparisons, we first argue that data redundancy is extremely severe in distillation (Sec. 3.1). Then, we model the dynamics of dataset distillation to explain our observations (Sec. 3.2).

### 3.1 DATA REDUNDANCY IN DATASET DISTILLATION

Dataset distillation learns a small synthetic dataset from the larger real dataset. To support the general the observation that a real dataset is redundant for the distillation task, we first conduct an ablation study on real datasets by **randomly** removing a portion of real data before the distillation process. We show an example in Fig. 2 on CIFAR10 with DC (Zhao et al., 2020) algorithm and IPC=1. After randomly dropping real data, the distillation performance remains stable until the drop rate is over 90%. With some other selection criteria (*e.g.*, the classification loss value of each sample), the dropping rate can be further increased to **99.96%**, *i.e.*, 20 samples are sufficient for comparable accuracy. Removing some samples even enhances the distillation accuracy. This simple experiment indicates that real data can be extremely redundant for dataset distillation, where some samples may be even "harmful".

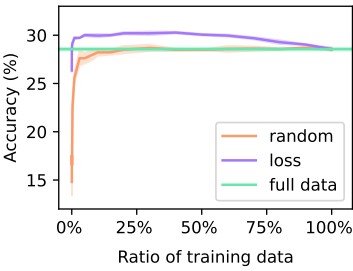

Figure 2: Example of distillation accuracy at different training data sizes. Orange: random dropping real data; Purple: dropping with an optimized criterion. Shaded region: standard deviation.

For further discussion, we first introduce key definitions pertinent to *critical samples*:

**Definition 1.** Given a real dataset $\mathcal{D}$ with size $m$, for a data selection policy $\pi$, we say $\Gamma(\pi) \in \mathbb{N}$ is the *critical sample size* if $\Gamma(\pi)$ is the minimal size *s.t.* the $\Gamma(\pi)$ samples in $\mathcal{D}$ selected by the policy $\pi$ is weakly $\epsilon$-close to the full dataset $\mathcal{D}$. The *critical sample ratio* $\gamma(\pi) = \Gamma(\pi)/m$.

Here we borrow the definition of $\epsilon$-close (Nguyen et al., 2020): two datasets are weakly $\epsilon$-close if the models trained on each dataset are close in terms of the empirical loss value. We use the 0-1 loss

Table 1: Comparison of $\gamma(rand)$ among datasets, distillation algorithms, and synthetic data sizes, *i.e.*, the **minimal** data ratio for comparable distillation accuracy.

| Dataset | CIFAR10 | | | CIFAR100 | | SVHN | | MNIST | | TinyImageNet | |
|---|---|---|---|---|---|---|---|---|---|---|---|
| IPC | 1 | 10 | 50 | 1 | 10 | 1 | 10 | 1 | 10 | 1 | 10 |
| DC | 10% | 30% | 50% | 50% | 50% | 40% | 20% | 1% | 3% | 60% | 50% |
| DSA | 15% | 30% | 30% | 60% | 60% | 40% | 30% | 70% | - | | |
| DM | 15% | 40% | 50% | 30% | 50% | 15% | 5% | 3% | 3% | 40% | 50% |
| MTT | 40% | 90% | 80% | 80% | 90% | 10% | 20% | 40% | 40% | 50% | - |

Table 2: $\gamma(rand)$ on more distillation algorithms.

| Dataset | IPC | CAFE | LinBa | IDC |
|---|---|---|---|---|
| CIFAR10 | 1 | 15% | 70% | 50% |
| | 10 | 11% | 30% | 10% |
| SVHN | 1 | 30% | 50% | 60% |
| | 10 | 60% | 60% | 30% |
| MNIST | 1 | 10% | 30% | 0.5% |
| | 10 | 1% | 40% | 40% |

Table 3: $\gamma(rand)$ among various inits on CIFAR10.

| IPC | Init | DC | DM |
|---|---|---|---|
| 1 | noise | 10% | 10% |
| | real | 10% | 15% |
| | herd | 10% | 15% |
| 10 | noise | 30% | 30% |
| | real | 30% | 40% |
| | herd | 30% | 30% |

Table 4: $\gamma(rand)$ among various networks on CIFAR10.

| IPC | Net | DC | DM |
|---|---|---|---|
| 1 | Conv | 10% | 15% |
| | MLP | 3% | 5% |
| | ResNet | 5% | 15% |
| | VGG | 10% | 5% |
| | AlexNet | 5% | 5% |
| 10 | Conv | 30% | 40% |
| | MLP | 40% | 40% |

function in the following discussion, so two datasets are weakly $\epsilon$-close means the models trained on them have similar distillation accuracy.

Therefore, the critical sample ratio is the minimal data ratio that can preserve the overall distillation performance with respect to a certain data selection policy, *e.g.*, the experiment in Fig. 2 indicates $\gamma(\pi) = 0.1$ when $\pi = $ random. A *smaller* critical sample ratio indicates *more* redundancy in the given real dataset and we have more freedom to prune the data. Note that $\gamma(\pi)$ may vary among different real and synthetic dataset settings and distillation algorithm settings.

Then we conduct more comprehensive comparison experiments on various datasets (Krizhevsky et al., 2009; Netzer et al., 2011; LeCun et al., 1998; Le & Yang, 2015), networks, distillation algorithms (Zhao et al., 2020; Zhao & Bilen, 2021; 2023; Cazenavette et al., 2022; Wang et al., 2022; Deng & Russakovsky, 2022; Kim et al., 2022b), initialization methods, and synthetic data sizes (represented by instance-per-class, IPC). We randomly drop some portions of the real data to find their critical sample ratio under random data selection ($\gamma(rand)$). We take the mean and standard deviation of the accuracy of 5 random removal trials. We use the default parameters of each method, which are detailed in Appendix Sec. F. The results are shown in Tab. 1, 2, 3, 4, which show that severe data redundancy widely exists in various dataset distillation settings. Some experiments on TinyImageNet at high IPC are not reported (marked with "-") due to its high computation cost and MTT exceeds our GPU memory limit. In most datasets and algorithms, less than 30% samples are sufficient for dataset distillation. We also observe some basic patterns:

1. Larger synthetic data size leads to a larger critical sample ratio.
2. Better distillation algorithms have larger critical sample ratios, *e.g.*, MTT performs better than DC and has larger critical sample ratios.
3. Initialization and architecture have only a minor influence on the critical sample ratio.

These observations indicate that we can only use a small subset of real data samples to reduce the training cost and to enable the optimization of the dataset distillation workflow from the aspect of real data utilization. Consequently, the goal of data-efficient distillation is **how to find the optimal data selection policy $\pi$ to minimize the critical sample ratio $\gamma(\pi)$**, or formally define:

**Definition 2.** A data selection policy is an *optimal selection* $\pi^*$ iff: $\pi^* = \arg\min_\pi \gamma(\pi)$.

### 3.2 THE INFORMATION TRANSPORT PERSPECTIVE OF DATASET DISTILLATION

To support the analysis of optimal data selection policy $\pi^*$, we first revisit the dataset distillation inspired by the information theory. The dataset distillation workflow can be regarded as the transport of "knowledge" from real data to synthetic data and, finally, to the classification model during training. We informally define the "knowledge" contained in a set of data samples $\mathcal{D}$ as their **utility**, denoted as $U(\mathcal{D})$. The samples with larger utility are more *valuable* for the learning.

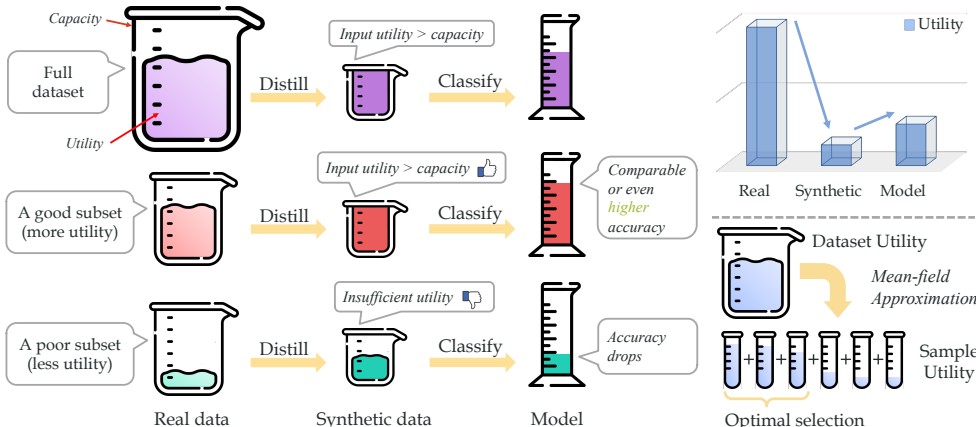

Figure 3: Data utility. The small capacity of synthetic data leads to a "bottleneck" of utility (right top). Thus, we seek an optimal selection policy that reduces real data while maintaining sufficient data utility, hence preserving or even enhancing the distillation performance. The right bottom figure shows the mean-field view of dataset utility and the corresponding greedy data selection strategy.

During the training process, the utility capacity of the classification model is limited, with the full data usually exceeding the capacity of the model. This gives rise to the popular data selection and distillation methods as minor data reduction will not degrade model performance. When we further consider the distillation step, the capacity of synthetic data is also limited and far smaller than that of full data. The size of synthetic data is often less than 1% of the whole dataset. Hence the utility flow from real data to synthetic data to the classification model forms a "bottleneck" structure, as shown in Fig. 3. In this case, dataset distillation has much more potential for data selection and data curation, and it is feasible and critical to analyze the critical samples in the dataset.

We denote the maximum utility that synthetic data can learn as $U_{syn}$. Once the real data utility exceeds $U_{syn}$, the distillation performance is maintained. Thus, we can define the optimal selection policy more specifically (card$(\cdot)$ measures the cardinality of a set):

**Definition 3.** Optimal data selection is to find $\mathcal{X}^* = \arg\min_{\mathcal{X} \subset \mathcal{D}} \text{card}(\mathcal{X}), \;\; s.t. \;\; U(\mathcal{X}) \geq U_{syn}.$

The modeling of dataset distillation also explains the rules in the previous ablation experiments:

1. A Larger synthetic set has larger $U_{syn}$, so more data are required to fill the utility capacity.
2. The synthetic data can receive more "knowledge" when using a superior distillation algorithm, *i.e.*, $U_{syn}$ is increased, so the critical sample ratio is increased.
3. The network architecture and initialization methods affect neither samples' utility nor $U_{syn}$, so they have a minor contribution to the critical sample ratio.

# 4 ESTIMATING DATA UTILITY

## 4.1 MEAN-FIELD VIEW OF DATA UTILITY

However, the above optimal selection is intractable and hard to optimize. To address our problem, we borrow the idea of mean-field theory from physics, which has been widely adopted in machine learning fields (Ranganath et al., 2014). Mean-field theory is a mathematical approach to understanding the behavior of large and complex systems with many interacting parts. The theory simplifies a complex system by assuming that each part of the system is only influenced by the average behavior of the other parts so that individual parts can be approximated by simpler independent systems.

In the case of data utility, we apply a mean-field approximation to decompose the utility of a set of data $U(\mathcal{D})$ into the combination of the utility of individuals $u(x_i)$:

$$U(\mathcal{D}) = U(\{x_1, x_2, \cdots, x_m\}) = u(x_1) + u(x_2) + \cdots + u(x_m). \tag{1}$$

Although the approximation omits the *high-order interaction* of utility between multiple samples, the mean-field family of $U(\mathcal{D})$ can greatly simplify the optimal selection policy: we can greedily

select samples with the largest utility value until the total utility exceeds $U_{syn}$ (Fig. 3). In the following sections, we will provide a comprehensive study of various indicators that are correlated with and can reflect the utility of data samples. These indicators will be employed in the data selection policy. More complex factors involving high-order interaction will be discussed in Sec. 4.4.

## 4.2 MEAN-FIELD INDICATORS OF DATA UTILITY

In this section, we present the data utility indicators under the mean-field view, *i.e.*, each sample is assigned a scalar that is positively related to its utility, followed by a thorough empirical comparison.

**a) Heuristic Indicators.** We first describe some heuristic data utility indicators, which reflect the data distribution and their training properties. The implementations are available in Appendix Sec. F.4. **(1) Classification Loss** indicates the learning difficulty of samples and distinguishes hard and easy cases. We train the classification model for 50~100 epochs for 50 experiment trials and take the average loss value of the *last* 10 epochs. **(2) Data density** around a sample depicts its scarcity and uniqueness. We first train the feature extractor on the dataset and extract its feature vectors. For each sample, we compute the mean distance to its K-nearest-neighbors in the same class. A negative sign is put for a density estimator. **(3) Distance to the cluster center**. Similar to (Liu et al., 2023), we adopt K-Means to explore the sub-cluster structure for each class. For each sample, we compute its distance to the nearest cluster center. The sub-cluster number is set to 128 by default. **(4) Distance to the initial image**. The utility of real samples may differ due to the initialization of synthetic data, *e.g.*, the initial images for real initialization may have less value since they are already the synthetic data themselves. We measure the L2 distance of each sample to the initial synthetic data with the same feature extractors in the density estimation.

**b) Monte-Carlo Indicator.** More directly, we propose a Monte-Carlo method to estimate the sample utility from the distillation accuracy. In the analysis in Sec. 3.2, we argue that if the data utility $U(\mathcal{X})$ is less than the capacity $U_{syn}$, the performance will drop. That is, if the data utility is small, the distillation accuracy can reflect the utility value and is an ultimate indicator of utility. Thus, we can estimate the sample utility with a randomized algorithm. We first select a data size $M$ which is less than $\Gamma(rand)$ to ensure $U(\mathcal{X}) < U_{syn}$. We repeat multiple trials (1000 trials by default) of random sampling a subset $\mathcal{X}_j$ with size $M$ from the real dataset and use the distillation accuracy $s_j$ as an indicator of the utility $U(\mathcal{X}_j)$. If the $i^{\text{th}}$ sample $x_i$ is visited in $T$ trials, its utility can be estimated as the average accuracy of these trials (Eq. 2). Considering both the mean-field assumption and the law of large numbers, the indicator is linear to the intrinsic sample utility $u(x_i)$ when $T$ is large:

$$\tilde{u}(x_i) = \frac{1}{T} \sum_{j:\mathcal{X}_j \ni x_i} s_j = \frac{1}{T} \sum_{j:\mathcal{X}_j \ni x_i} U(X_j) \xrightarrow{T \to \infty} \frac{N-M}{N-1} u(x_i) + \frac{(M-1)N}{N-1} \mathbb{E}(u(x)), \quad (2)$$

since the second term is a constant and $(N-M)/(N-1) > 0$. We prove Eq. 2 in the appendix.

**Comparison of Utility Indicators.** We compare the above utility indicators via two metrics. Appendix Sec. B gives the full experiment results and extra comparison to coreset selection algorithms.

**I. Stratified analysis of utility.** We sorted the samples by their utility and split the dataset into a few subgroups. Their utility values are monotonously increasing. By conducting distillation on each group, we can evaluate whether the utility indicator induces an ideal total ordering for samples.

First, we identify the direction or "polarity" of the indicators based on the results, *e.g.*, which samples have larger utility, those with a larger or smaller loss value. We find that the samples with larger utility are those with 1) small loss value, 2) large density, 3) closer to cluster centers, 4) closer to initialization samples, or 5) larger Monte-Carlo utility. This indicates the distillation prefers common samples that are easier to learn rather than corner cases and outliers, which is consistent with our model in Sec. 3.2: due to the small capacity of synthetic data, the algorithms tend to learn the easy samples and common patterns of the real dataset.

Second, we compare the proposed indicators. Most indicators can correctly distinguish the samples with high or low utility, except the density and initialization distance which cannot accurately find the large utility samples. The loss value and Monte-Carlo estimator are significantly superior to the rest since they have a better discriminating ability on both high and low-utility samples.

**II. Critical sample ratio comparison.** We use the utility estimation to greedily select samples, and compare their critical sample ratio $\gamma(\pi)$ in various settings. Among the utility indicators, classifi-

Table 5: Comparison of $\gamma(loss)$ among datasets, distillation algorithms, and synthetic data sizes, *i.e.*, the **minimal** data ratio for comparable distillation accuracy

| Dataset | CIFAR10 | | | CIFAR100 | | SVHN | | TinyImageNet | |
|---------|------|------|------|------|------|------|------|------|------|
| IPC | 1 | 10 | 50 | 1 | 10 | 1 | 10 | 1 | 10 |
| DC | 0.5% | 70% | 30% | 5% | 10% | 20% | 80% | 3% | 20% |
| DSA | 0.5% | 40% | 15% | 3% | 20% | 5% | 40% | 1% | - |
| DM | 0.5% | 50% | 60% | 0.5% | 10% | 1% | 15% | 0.5% | 3% |
| MTT | 40% | 80% | 80% | 40% | 80% | 1% | 5% | 30% | - |

Table 6: Comparison of $\gamma(loss)$ on more distillation algorithms.

| Dataset | IPC | CAFE | LinBa | IDC |
|---------|-----|------|-------|-----|
| CIFAR10 | 1 | 0.2% | 40% | 15% |
| | 10 | 11% | 20% | 5% |
| SVHN | 1 | 40% | 80% | 70% |
| | 10 | 1% | 70% | 80% |

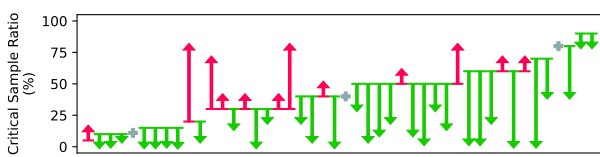

Figure 4: The comparison of $\gamma(rand)$ and $\gamma(loss)$: the arrows point from $\gamma(rand)$ to $\gamma(loss)$. Green: selection by loss is superior; Red: selection by loss is inferior to random.

cation loss and the Monte-Carlo method significantly reduce the critical sample ratio to **lower than 1%**, *i.e.*, the samples selected by them are more critical for dataset distillation. Despite their comparable performance, the Monte-Carlo method is far more inefficient and less applicable due to its slow convergence: *e.g.*, for CIFAR10, we distill from 5% of the real data 1000 times, which takes over 100x the time of distillation itself. Thus, we argue that **the loss value is the most effective and efficient data utility indicator**, which will be utilized and discussed in the following sections. Among the other criteria, the data density and clustering methods also perform above random selection, though inferior to loss and Monte Carlo, showing that data distribution clues also help find valuable samples. Whilst, the initialization distance does not give a meaningful sample ordering.

## 4.3 EXTENDED DISCUSSION ON THE LOSS INDICATOR

Considering both the performance and the computation complexity, the loss value is a better estimator for data utility. Thus we propose to use a loss utility indicator in the greedy selection method. We conducted an in-depth study on loss value and have the following observations.

**Dropping by loss consistently outperforms random selection.** We extend the experiments of critical sample ratio $\gamma(loss)$ with loss utility indicator on 4 datasets, 7 algorithms, and 3 IPC settings as shown in Tab. 5 and 6. Experiments on more SOTA algorithms can be found in Appendix Sec. B.3. We also show the comparison between random selection and selection with loss in Fig. 4. The greedy selection with loss consistently outperforms random selection. In most cases, the critical sample ratios are $< 10\%$, significantly reducing the training cost. We also find dropping data does not drop the transferability in Appendix Sec. D.

**Dropping by loss can enhance distillation.** Surprisingly, we also observe that dropping real data can improve performance. We show the best performance during data dropping in Tab. 7 on DC, DSA, DM, and MTT. In almost all cases, the data selection can notably promote distillation accuracy. This may imply some samples have *negative utility* that are detrimental to distillation. With accurate utility estimation, greedy selection can cancel the these negative impacts. This observation inspires new approaches that leverage the data utility and exploit all data samples in different quality, enabling future analysis of network dynamics and dataset distillation. Appendix Sec. B.3 and C show more support for SOTA methods and an explanation from the variance/outlier perspective.

**Early epoch loss in very few trials is also accurate for selection.** The twig is bent, and so is the tree inclined. We find that the loss values in very early classification epochs are informative enough for data selection, probably as the early dynamics of samples can reflect their training difficulty and importance. Moreover, the number of trials to obtain loss values has little influence on the selection performance, as shown by the empirical results on CIFAR10, IPC=1, and DC. We take the average loss curves of 1, 5, 10, and 50 trials of classification, and take the mean loss value for the first 1, 5, 10, and 100 epochs when the training converges at around 100 epochs. In Fig. 5, we use the same stratification method as Sec. 4.1 to evaluate the sample ordering induced by the loss values

Table 7: Best performance w/ data dropping. The performance difference between the *full dataset* and *x% of real data* are shown in parentheses (†: compare to our reproduced accuracy).

| Dataset | IPC | DC | DSA | DM | MTT |
|---|---|---|---|---|---|
| CIFAR10 | 1 | 30.0±0.1 (**+1.7**, 5%) | 30.9±0.1 (**+2.6**, 20%) | 29.7±0.3 (**+3.7**, 5%) | 46.3±0.8 (**+0.2**, 80%) |
| | 10 | 44.9±0.4 (**+0.0**, 70%) | 52.4±0.2 (**+0.2**, 50%) | 50.0±0.2 (**+1.1**, 50%) | 65.7±0.3 (**+0.4**, 90%) |
| | 50 | 54.9±0.5 (**+1.0**, 60%) | 61.5±0.7 (**+0.9**, 20%) | 63.4±0.2 (**+0.4**, 60%) | 72.0±0.4 (**+0.4**, 95%) |
| CIFAR100 | 1 | 14.1±0.1 (**+1.3**, 20%) | 15.6±0.1 (**+1.7**, 20%) | 14.9±0.5 (**+3.5**, 10%) | 24.6±0. (**+0.3**, 90%) |
| | 10 | 26.5±0.3 (**+1.3**, 30%) | 32.5±0.4 (**+0.2**, 30%) | 32.4±0.3 (**+2.7**, 50%) | 40.1±0.5 (**+0.0**, 80%) |
| SVHN | 1 | 32.2±0.5 (**+1.0**, 20%) | 28.9±1.3 (**+0.1**, 40%) | 29.8±0.5 (**+6.3**†, 20%) | 43.0±1.1 (**+3.2**†, 10%) |
| | 10 | 76.2±0.6 (**+0.1**, 40%) | 80.0±0.8 (**+0.8**, 70%) | 74.6±0.3 (**+0.9**†, 30%) | 78.1±0.5 (**+0.9**†, 20%) |
| TinyImageNet | 1 | 4.9±0.1 (**+0.2**†, 30%) | 4.3±0.0 (**+0.6**†, 5%) | 4.8±0.1 (**+0.9**, 10%) | 9.0±0.4 (**+0.2**, 10%) |
| | 10 | 12.8±0.0 (**+0.2**†, 20%) | - | 17.5±0.1 (**+4.6**, 30%) | - |

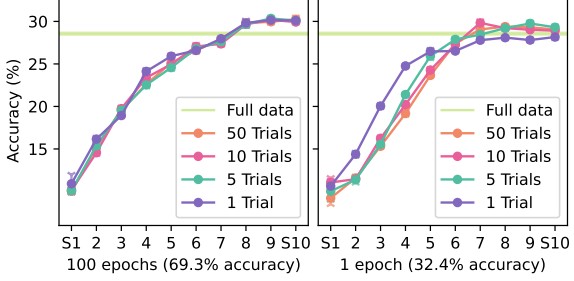

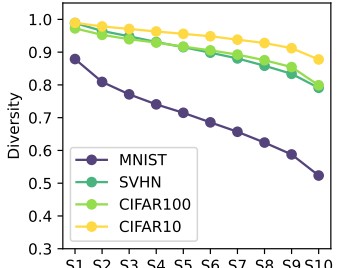

Figure 5: Stratified distillation performance of different classification trials and epochs numbers. An ideal stratification yields a monotonously increasing performance curve.

Figure 6: The diversity of each stratified subgroup. MNIST significantly suffers from diversity vanishing.

(Appendix Fig. 7 for the full figure). Tab. 8 lists the corresponding critical sample ratios. Most loss values produce a good stratification and thus have very small $\gamma$, the best of which decrease $\gamma$ to 0.04%, *i.e.*, 2 samples per class are enough for comparable performance with this selection criterion. Even in some extreme settings, *e.g.*, 5 trials of 1 epoch training, they are sufficient for an informative loss value. Only extreme cases such as 1 epoch and 1 trial show degradation. The two observations can be utilized to substantially reduce the computational burden of generating loss values, which now can be ignored or naturally embedded in the distillation process itself, extending our paradigm to broader applications.

Table 8: Comparison of $\gamma(loss)$ of difference classification trial and epochs.

| Trial Number | 1 | 5 | 10 | 50 |
|---|---|---|---|---|
| 1 Epoch | 10% | 0.3% | 0.2% | 0.1% |
| 5 Epochs | 0.2% | 0.04% | 0.1% | 0.1% |
| 10 Epochs | 0.2% | 0.04% | 0.04% | 0.04% |
| 100 Epochs | 0.2% | 0.1% | 0.2% | 0.1% |

## 4.4 HIGHER-ORDER INTERACTION OF DATA UTILITY

In Sec. 4.1, we leverage mean-field approximation to simplify the data utility. The experiments and discussion above also justify this simplification. However, the high-order information and interactions between samples do exist, and in some extreme scenarios, these interactions are not ignorable. For example, *data diversity* is a higher-order data utility indicator since it is a property of the population rather than an individual. We observe diversity affects data distillation. The data utility paradigm always performs poorly on the MNIST dataset. We conduct stratified experiments on MNIST with loss value indicator and observe that both the subgroups with large loss ($S_1$) and small loss ($S_{10}$) have lower accuracy, which is due to the *diversity vanishing* in the small loss subgroups. We use the quotient of intraclass variance and interclass variance as the diversity metric (a large value indicates larger diversity). As shown in Fig. 6, only MNIST severely drops the diversity for the subgroups with a small loss. This phenomenon also exists in other utility indicators such as data density, which spoils the data utility paradigm. This phenomenon reflects the impact of high-order interaction of data utility. It can be challenging to incorporate diversity into consideration without sacrificing the efficiency in our paradigm (*e.g.*, leveraging annealing or Monte-Carlo algorithms) so we leave it to future work. In-depth discussion and modeling of high-order data utility involve complex systems which is beyond our scope. It is worth noting that the impact of diversity vanishing can be negligible in most realistic scenarios (*e.g.*, CIFAR), especially for large-scale datasets due to their large overall diversity. We provide the stratified visualization in Appendix Sec. G.2.2.

Table 9: Performance of state-of-the-art and our runtime pruning approach (†: reproduced accuracy).

| Dataset | IPC | DSA | DM | MTT | DC | IDC | DC+Ours | IDC+Ours | Full dataset |
|---------|-----|-----|-----|-----|-----|-----|---------|----------|--------------|
| CIFAR10 | 1 | 28.8±0.7 | 26.0±0.8 | 46.3±0.8 | 28.3±0.5 | 55.3±0.3† | 29.3±0.1 | **56.1±0.3** | 84.8±0.1 |
|  | 10 | 52.1±0.5 | 49.9±0.6 | 65.6±0.7 | 44.9±0.5 | 65.0±0.5† | 45.1±0.4 | **65.3±0.1** |  |
| CIFAR100 | 1 | 13.9±0.3 | 11.4±0.3 | 24.3±0.3 | 12.8±0.3 | 31.3±0.2† | 13.2±0.4 | **32.1±0.2** | 56.2±0.3 |
|  | 10 | 32.3±0.3 | 29.7±0.3 | 40.1±0.4 | 25.2±0.3 | 40.8±0.3† | 25.9±0.5 | **41.3±0.4** |  |
| SVHN | 1 | 27.5±1.4 | - | - | 31.2±1.4 | 68.7±0.6† | 31.7±0.7 | **70.0±0.7** | 95.4±0.1 |
|  | 10 | 79.2±0.5 | - | - | 76.1±0.6 | 82.7±0.6† | 76.6±0.4 | **83.0±0.2** |  |

Table 10: Dataset distillation on large-scale image and video datasets (*est.* = estimated).

| Dataset | IPC | Algorithm | Distill Full Data | | Distill with Selection (Ours) | | Random Real |
|---------|-----|-----------|----------|---------------|----------|---------------|-------------|
|  |  |  | Accuracy | Training Time | Accuracy | Training Time |  |
| ImageNet-1K (1.3M images) | 1 | DC | 1.79±0.04 | 23.6h | **1.93±0.06** | **17.2h** | 0.43±0.02 |
|  |  | DM | 1.58±0.11 | 22.4h | **1.82±0.14** | **9.8h** |  |
|  |  | MTT | - | 205.0h (*est.*) | **1.97±0.04** | **31.0h** |  |
|  | 10 | DM | 3.86±0.16 | 24.7h | **5.11±0.08** | **15.1h** | 1.57±0.21 |
|  | 50 | DM | 8.22±0.86 | 35.0h | **9.23±0.40** | **20.4h** | 5.29±0.70 |
| Kinetics-400 (300K videos) | 1 | DM | 2.78±0.14 | 37.3h | **2.88±0.12** | **29.4h** | 0.90±0.23 |
|  |  | MTT | - | 460.8h (*est.*) | **2.69±0.17** | **76.6h** |  |
|  | 10 | DM | 9.48±0.15 | 43.5h | **9.56±0.08** | **32.1h** | 3.33±0.43 |

## 5 HARNESSING DATA UTILITY

### 5.1 A SIMPLE CASE TO EXTEND GRADIENT MATCHING METHODS

Based on the data utility, we propose a baseline to leverage utility during run time, *i.e.*, *no pre-computing of data utility*. We present a simple but effective plug-and-play mechanism for *gradient-matching-based* methods with only 3 lines extra code (Appendix Sec. E), to exploit the bias towards small values: dropping the samples with larger losses in each batch. Since the loss has been computed during gradient matching, the strategy brings no computational overhead. Rather, it reduces the samples hence reducing the backpropagation time.

We apply this approach to DC and IDC (Kim et al., 2022b). The default pruning rate is 30% (Appendix Sec. F for details). Shown in Tab. 9, our simple strategy can significantly enhance the current distillation algorithms by 1% with efficiency. This experiment shows the feasibility of embedding the data utility mechanism into the current distillation paradigm to boost performance. The data utility exhibits great potential for practical applications and is promising for future extensions.

### 5.2 EFFICIENT DISTILLATION OF LARGE-SCALE DATASETS

With our data utility paradigm, the existing algorithms can be efficiently extended to larger-scale and more heterogeneous datasets. We apply the data utility selection to the distillation of ImageNet-1K (Deng et al., 2009) and scale up to IPC=50, and for the very first time, we distill large-scale video dataset Kinetics-400 (Carreira & Zisserman, 2017) (detailed in Appendix Sec. F). The results are listed in Tab. 10. Most methods struggle with high IPC due to demanding GRAM, except DM which allows class-separate training. MTT is extremely expensive for large-scale data due to its expert training. However, our method effectively mitigates training costs. The data utility paradigm significantly reduces the training time by at most 60%, while maintaining or enhancing the performance. Our paradigm is especially suitable for large-scale scenarios when the data size increases towards infinite, whose signal-to-noise ratio continues to decrease. Our method can discover the critical samples and dispose of the noise. We will extend to distill even larger datasets in the future.

## 6 CONCLUSION

This work presents a new perspective on dataset distillation, contributing to the very first principled work on data redundancy in dataset distillation. We model the data utility to identify the sample value, whose efficacy is validated by extensive experiments. Further, we design a baseline to leverage the data utility paradigm which outperforms the current algorithms. With data utility selection, we can efficiently distill larger and more heterogeneous datasets like ImageNet and Kinetics. Our paradigm will bring inspiration and spawn more fruitful work toward efficient dataset distillation.

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

We provide the following details and analyses in the appendix:

Sec. A: The Convergence of The Monte-Carlo Indicator.

Sec. B: More Experimental Results for Utility Indicators.

Sec. C: Variance and Outlier Analysis.

Sec. D: Cross-architecture Generalization.

Sec. E: Pseudocode of the Proposed Baseline.

Sec. F: Implementation Details.

Sec. G: More Visualizations.

Sec. H: Limitations.

Sec. I: Licenses.

## A  THE CONVERGENCE OF THE MONTE-CARLO INDICATOR

As introduced in the main paper, the Monte-Carlo indicator is linear to the intrinsic sample utility when the number of randomized trials is large. The proposition can be explained intuitively: if a sample achieves high accuracy with any other groups of samples, then this sample must be important. We also provide detailed proof below.

**Notations**: The full real dataset $\mathcal{D}$ has $N$ samples. The randomized experiment is repeated for $K$ times and we randomly draw a subset $\mathcal{X} \subset \mathcal{D}$ with $M$ samples each time. We use the distillation accuracy $s$ to represent the utility of the subset $U(\mathcal{X})$. Based on this information, we hope to estimate the utility of the individual $u(x_i)$, $\forall i = 1, 2, ..., N$ and assume that sample $x_i$ appears $T$ times among the $K$ random trials.

**Definition**: The Monte-Carlo indicator for sample $x_i$ is the average accuracy of all randomized trials that involve $x_i$: $\tilde{u}(x_i) = \frac{1}{T} \sum_{j:\mathcal{X}_j \ni x_i} s_j$.

**Proposition**: The Monte-Carlo indicator converges to a value that is linear to the intrinsic sample utility as $T \to \infty$.

**Proof**:

$$
\begin{aligned}
\tilde{u}(x_i) &= \frac{1}{T} \sum_{j:\mathcal{X}_j \ni x_i} s_j = \frac{1}{T} \sum_{j:\mathcal{X}_j \ni x_i} U(X_j) \\
&= u(x_i) + \frac{1}{T} \sum_{j:\mathcal{X}_j \ni x_i} U(X_j - \{x_i\}) \\
&\xrightarrow{T \to \infty} u(x_i) + \mathbb{E}_{\mathcal{X}}[U(\mathcal{X})], \quad s.t. \ \mathcal{X} \subset \mathcal{D} - \{x_i\}, \mathrm{card}(\mathcal{X}) = M - 1 \\
&= u(x_i) + \mathbb{E}_{\mathcal{X}}\left[\sum_{x \in \mathcal{X}} u(x)\right] \\
&= u(x_i) + (M-1)\mathbb{E}_{\mathcal{X}}\mathbb{E}_x[u(x)|\mathcal{X}].
\end{aligned}
\tag{3}
$$

Applying *the law of total expectation*, or namely *the tower rule*, we have:

$$
\begin{aligned}
& u\left(x_i\right) + (M-1)\mathbb{E}_{\mathcal{X}}\left[\mathbb{E}_x\left(u(x)|\mathcal{X}\right)\right] \\
=& u\left(x_i\right) + (M-1)\mathbb{E}_{x\in\mathcal{D}-\{x_i\}}\left[u(x)\right] \\
=& u\left(x_i\right) + (M-1)\left[\frac{1}{N-1}\sum_{x\in\mathcal{D}-\{x_i\}} u(x)\right] \\
=& u\left(x_i\right) + (M-1)\left[\frac{1}{N-1}\sum_{x\in\mathcal{D}-\{x_i\}} u(x)\right] \\
=& u\left(x_i\right) + (M-1)\left[\frac{1}{N-1}\left(-u(x_i)+\sum_{x\in\mathcal{D}} u(x)\right)\right] \\
=& \frac{N-M}{N-1} u\left(x_i\right) + (M-1)\frac{N}{N-1}\mathbb{E}_{x\in\mathcal{D}}\left(u(x)\right),
\end{aligned}
\tag{4}
$$

where $(N-M)/(N-1) > 0$ and the second term is constant w.r.t $x_i$. Therefore, $\tilde{u}(x_i)$ is positively linear related to $u(x_i)$.

$\square$

However, the convergence speed can be slow. Note that we draw $M$ samples from the population with $N$ samples for $K$ trials, so for each sample, the average number of its occurrences is $\mathbb{E}(T) = MK/N$ and we hope this value to be large enough according to the theorem of the large number. Since the choice of $M$ should satisfy $U(\mathcal{X}) < U_{syn}$, the sampling ratio $M/N$ can be small, hence a larger $K$ is required which slows the convergence, *e.g.*,, for CIFAR10 dataset and DC algorithm, the critical sample ratio $\gamma(rand) = 0.1$ so we use a smaller sample ratio $M/N = 0.05$ for $U(\mathcal{X}) < U_{syn}$. If we want the samples to have $T = 100$ on average, we should conduct $K = 2000$ distillation experiments, which are far more expensive than the distillation itself. So it is another "*penny-wise and pound-foolish*" issue (refer to the second paragraph in Sec. 1 in the main paper) to use the Monte-Carlo indicator for data selection. However, considering its stable performance, we hope for a more efficient Monte Carlo algorithm in the future.

## B    More Experimental Results for Utility Indicators

### B.1    Comparison of Utility Indicators

In Sec. 4.2 in the main paper, we propose various utility indicators and give our comparison conclusions. Here we provide the full details and results of comparisons of the proposed utility indicators. We use two metrics to compare the estimation.

#### B.1.1    Stratified Analysis of Utility

We first inspect the utility indicators by stratification. We sort the samples according to their utility and split the dataset into a few subgroups and their utility values are monotonously increasing. By conducting distillation on each group, we can evaluate whether the utility estimation induces an ideal total ordering for the greedy selection method. The stratified analysis also helps determine the direction or "polarity" of the indicators, *e.g.*, which samples have larger utility, those with a larger loss or those with a smaller loss.

We conduct experiments on CIFAR10 and with DC (Zhao et al., 2020) algorithm and IPC=1. The dataset is stratified according to the utility indicator values to 10 subgroups, denoted as $S_1, S_2, \cdots, S_{10}$. Note that we sort and split the dataset class-by-class to avoid imbalancedness. The results are shown in Tab. 11.

We first observe the polarity of the indicators: the samples that have larger utility are those with 1) small loss value, 2) large density, 3) closer to cluster centers, 4) closer to initialization samples, or 5) larger Monte-Carlo utility. This indicates the distillation prefers common samples that are easier to learn rather than corner cases and outliers, which is consistent with our model in Sec. 3.2 in the

Table 11: The stratified experiment of various utility indicators on CIFAR10 with DC (Zhao et al., 2020) and IPC=1. The indicators marked with $\searrow$ means they are sorted from large to small values (*e.g.*, for loss indicator, $S_1$ has the *largest* loss values and $S_{10}$ has the *smallest* values), and $\nearrow$ means they are sorted from small to large (Red: inferior to distillation on the full dataset; Blue: superior.)

| Indicator | $S_1$ | $S_2$ | $S_3$ | $S_4$ | $S_5$ | $S_6$ | $S_7$ | $S_8$ | $S_9$ | $S_{10}$ |
|---|---|---|---|---|---|---|---|---|---|---|
| Random | 28.3±0.5 | 28.5±0.5 | 28.4±0.3 | 28.2±0.3 | 27.9±0.2 | 28.8±0.3 | 27.9±0.3 | 27.8±0.4 | 28.5±0.5 | 27.9±0.3 |
| Loss ($\searrow$) | 13.0±0.1 | 18.4±0.1 | 22.9±0.1 | 24.9±0.1 | 26.6±0.1 | 27.6±0.1 | 27.5±0.2 | 28.9±0.2 | 30.0±0.2 | 29.8±0.2 |
| Density ($\nearrow$) | 17.2±0.1 | 20.7±0.1 | 23.0±0.1 | 25.2±0.2 | 26.9±0.2 | 27.1±0.2 | 27.9±0.1 | 28.2±0.1 | 28.6±0.1 | 27.4±0.2 |
| Cluster distance ($\searrow$) | 23.1±0.2 | 24.6±0.2 | 26.7±0.1 | 27.0±0.3 | 27.6±0.2 | 27.9±0.1 | 28.3±0.1 | 28.5±0.1 | 28.8±0.2 | 27.5±0.0 |
| Init distance ($\searrow$) | 20.6±1.1 | 24.0±0.1 | 25.3±0.2 | 26.2±0.5 | 26.9±0.9 | 27.4±1.0 | 27.5±1.0 | 27.6±0.2 | 24.9±0.9 | 21.2±2.3 |
| Monte-Carlo($\nearrow$) | 23.8±0.2 | 25.0±0.1 | 25.7±0.1 | 27.1±0.2 | 28.0±0.1 | 28.4±0.0 | 29.0±0.1 | 30.2±0.1 | 30.8±0.1 | 31.2±0.1 |

Table 12: Comparison of $\gamma(\pi)$ of the greedy sample method with various utility estimators, *i.e.*, the **minimal** data ratio for comparable distillation accuracy (**bold**: best, underline: second best).

| Algorithm | Dataset | IPC | Random | Loss | Density | Cluster Distance | Init Distance | Monte-Carlo |
|---|---|---|---|---|---|---|---|---|
| DC | CIFAR10 | 1 | 10% | **0.5%** | 15% | 15% | 90% | 1% |
| | | 10 | **30%** | 70% | 70% | 90% | 90% | 70% |
| | | 50 | 50% | **30%** | 80% | 70% | 90% | 80% |
| | CIFAR100 | 1 | 50% | **5%** | 20% | 40% | 80% | **5%** |
| | SVHN | 1 | 40% | 20% | 40% | 30% | 100% | **5%** |
| | MNIST | 1 | 1% | 70% | 50% | 40% | 90% | **0.3%** |
| DM | CIFAR10 | 1 | 15% | **0.5%** | 15% | 10% | 80% | 1% |
| DSA | CIFAR10 | 1 | 15% | **0.5%** | 15% | 15% | 60% | 1% |

main paper: since the capacity of synthetic data is limited and relatively small, the algorithm tends to learn the easy samples and common patterns of the real dataset.

We also compare the proposed indicators. Most indicators can identify the samples with high or low utility, *i.e.*, the performance is increasing from $S_1$ to $S_{10}$ and we can find the most valuable samples according to these indicators. Among these indicators, the density and initialization distance cannot accurately find the large utility samples (accuracy drops at $S_{10}$). Also, the loss value and Monte-Carlo estimator are significantly superior to the rest since their accuracy on $S_9$ and $S_{10}$ is higher, with their performance on $S_1$ and $S_2$ worse, showing that fewer valuable samples are misclassified to these subgroups.

### B.1.2 CRITICAL SAMPLE RATIO COMPARISON

We use the utility estimation to greedily select samples, and compare their critical sample ratio $\gamma(\pi)$ on different datasets, algorithms, and IPCs. We use the same experiment settings as Sec. 3.1 in the main paper, which is detailed in App. Sec. F.

The experiments are shown in Tab. 12. Among the proposed utility estimators, classification loss and the Monte-Carlo method significantly reduce the critical sample ratio, *i.e.*, the samples selected by these criteria are more critical for dataset distillation. Among the other criteria, the data density and clustering methods also perform above random selection, though inferior to loss and Monte Carlo, showing that data distribution clues also help find valuable samples. In comparison, the initialization distance does not give a meaningful ordering of the samples.

### B.2 COMPARISON TO CORESET SELECTION METHODS

The existing coreset selection methods can also be exploited as critical sample selection algorithms. So we conduct a comparison with recent coreset selection methods on CIFAR10 and DC algorithm, including GradMatch (Killamsetty et al., 2021a) and GLISTER (Killamsetty et al., 2021b). We use the algorithms implemented by the CORDS package. The results are shown in Tab. 13 and 14. GradMatch coreset selection can achieve a 3% critical sample ratio when IPC=1, though still worse than the loss indicator. The GLISTER algorithm does not perform well on dataset distillation and is worse than random selection. Thus, on the data pruning for dataset distillation, our loss indicator can surpass some sophisticated selection algorithms.

Table 13: Comparison to coreset selection on CIFAR10, IPC=1 and DC algorithm.

| Selection Criterion | Sample Ratio | | | | | | 100% (Full) |
|---|---|---|---|---|---|---|---|
| | 1% | 3% | 5% | 10% | 20% | 30% | |
| Random | 25.6±0.6 | 27.6±0.8 | 27.6±0.6 | 28.2±0.3 | 28.5±0.4 | 28.7±0.3 | |
| Loss (remove large) | **29.7±0.1** | **29.7±0.0** | **30.0±0.1** | **30.0±0.2** | **30.2±0.1** | **30.2±0.2** | 28.3±0.5 |
| Coreset: GradMatch | 26.5±0.5 | 28.0±0.3 | 28.7±0.5 | 28.4±0.3 | 28.9±0.2 | 29.4±0.3 | |
| Coreset: GLISTER | 23.8±0.5 | 26.9±0.2 | 23.5±0.1 | 21.3±0.6 | 24.0±0.9 | 24.9±0.8 | |

Table 14: Comparison to coreset selection on CIFAR10, IPC=50 and DC algorithm.

| Selection Criterion | Sample Ratio | | | | 100% (Full) |
|---|---|---|---|---|---|
| | 30% | 40% | 50% | 70% | |
| Random | 53.0±0.2 | 53.6±0.3 | 54.0±0.5 | 54.2±0.3 | |
| Loss (remove large) | **54.1±0.2** | **54.9±0.3** | **55.3±0.3** | **56.0±0.2** | 54.1±0.3 |
| Coreset: GradMatch | 49.0±0.3 | 49.1±0.3 | 49.1±0.4 | 49.1±0.5 | |
| Coreset: GLISTER | 41.9±0.4 | 43.9±0.5 | 44.0±0.5 | 47.0±0.5 | |

## B.3 MODE RESULTS FOR LOSS INDICATOR

Besides the basic evaluation of the loss indicator with DC, DM, DSA, MTT, $etc$, we apply data dropping to more algorithms in Tab. 15 and 16. The critical sample selection can prune up to 70% samples for the most recent distillation algorithms. And our technique could consistently enhance these algorithms with fewer training samples. The results of the experiments provide strong evidence for our conclusion of high redundancy and potential negative sample utility in the dataset distillation.

Table 15: Comparison of $\gamma(loss)$ on more distillation algorithms KIP, FRePo, HaBa, and RFAD, with IPC=1.

| Dataset | CIFAR10 | MNIST |
|---|---|---|
| KIP (NN) | 70% | 50% |
| FRePo | 80% | 60% |
| HaBa | 80% | 80% |
| RFAD | 30% | 70% |

Table 16: Best performance w/ data dropping for SOTA algorithms RFAD and IDM. The performance difference between the full dataset and x% of real data are shown in parentheses (†: compare to our reproduced accuracy).

| Dataset | IPC | RFAD | IDM |
|---|---|---|---|
| CIFAR10 | 1 | 53.7±0.9 (+0.1, 50%) | 46.3±0.4 (**+0.7**, 90%) |
| | 10 | 66.7±0.2 (**+0.4**, 95%) | 58.9±0.3 (**+0.3**, 95%) |
| | 50 | 71.9±0.2 (**+0.8**, 95%) | 67.8±0.2 (**+0.3**, 90%) |
| CIFAR100 | 1 | 30.4±0.6 (**+4.1**, 40%) | 25.9±0.3 (**+5.8**, 60%) |
| | 10 | 38.4±0.2 (**+6.4**, 40%) | 45.9±0.1 (**+0.8**, 60%) |

## B.4 STRATIFIED EXPERIMENTS FOR EXTENDED DISCUSSION OF LOSS INDICATOR

In Sec. 4.3 and Fig. 5 in the main paper, we discuss how the number of classification epochs and trials affect the performance of loss indicators. We provide the full experiment plots in Fig. 7, showing that we can obtain an accurate loss indicator by training very few epochs very few times. Only extreme cases such as 1 epoch and 1 trial show degradation.

## C VARIANCE AND OUTLIER ANALYSIS

For a more solid study on why the distillation performance can be enhanced by data pruning, we conduct data variance and outlier analysis at different sample ratios. As shown in Fig. 8, when we gradually prune the training samples, the number of outliers decreases, which helps the model learning. Concurrently, the variance and the data diversity also decrease, which harms model learning. The two factors compete during pruning. When the effect of outlier decrement becomes dominant, pruning data results in performance increment.

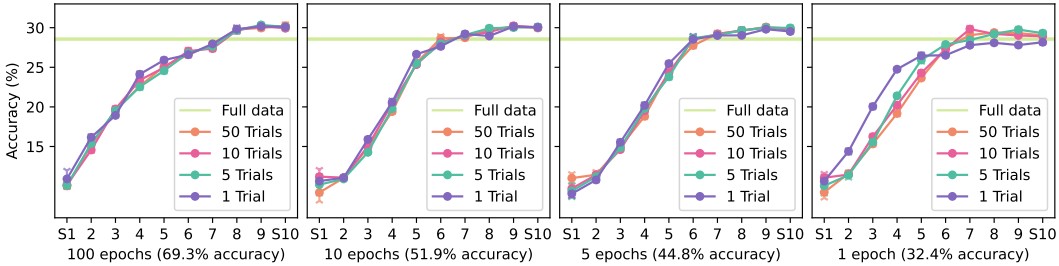

Figure 7: The stratified distillation performance of different classification trials and epochs numbers. The classification accuracies are shown in the figure titles. An ideal stratification can yield a monotonously increasing performance curve.

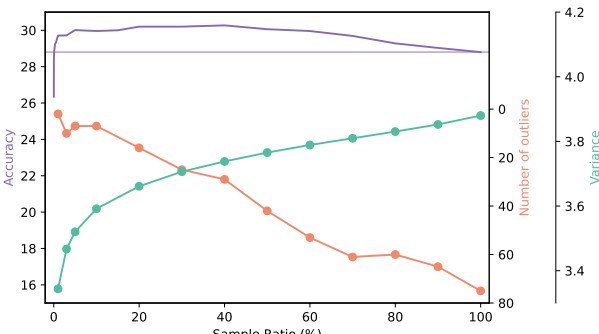

Figure 8: The data variance and number of outliers at different sample ratios according to the loss indicator. When we prune the data, the variance or diversity decreases, while the number of outliers also decreases. (The axis of the number of outliers is reversed since the fewer outliers the better for model learning.)

## D  CROSS-ARCHITECTURE GENERALIZATION

We conducted a cross-architecture evaluation to verify whether the data pruning harms the generalization ability. We first follow DC (Zhao et al., 2020) to experiment on MNIST and IPC=1. We remove the training samples with the largest loss values and we compare the training subsets with 100% (full data, original setting in DC paper), 10%, 5%, and 3%. The results are shown in Tab. 17 in the rebuttal PDF file, showing that pruning the training dataset does not damage the generalization ability of the distilled data. On the contrary, in most slots (28/36), data pruning can even enhance the generalization ability.

We also conduct experiments on larger IPCs with DC and MTT. The results are shown in Tab. 18 and  19. On larger IPCs, pruning the training dataset still does not damage the generalization ability of the distilled data.

## E  PSEUDOCODE OF THE PROPOSED BASELINE

The implementation of our proposed baseline method is simple and easy to plug in, as the following 3 lines of core pseudocode (Code 1).

Code 1: The pseudocode of real data gradient in runtime data pruning w/o extra overhead.

```
loss_real = cross_entropy_loss(image_real, label_real, reduction=None)
loss_real = loss_real.topk(batchsize - num_to_drop, largest=False).mean()
grad_real = gradient(loss_real, net_parameters)
```

Table 17: Cross-architecture generalization on MNIST and IPC=1 with DC algorithm.

| Evaluate Network | Sample Ratio | Distill Network | | | | | |
|---|---|---|---|---|---|---|---|
| | | MLP | ConvNet | LeNet | AlexNet | VGG | ResNet |
| MLP | 100% | **75.7±0.9** | 73.9±0.6 | **76.0±0.3** | **74.9±0.3** | 72.8±0.2 | **77.5±0.3** |
| | 10% | 74.8±0.5 | **74.8±0.4** | 73.3±0.6 | 73.4±0.4 | 71.8±0.2 | 74.9±0.3 |
| | 5% | 73.6±0.9 | 74.4±0.7 | 73.4±0.4 | 73.2±0.7 | 71.6±0.1 | 74.7±0.8 |
| | 3% | 73.3±0.3 | 74.7±0.3 | 72.9±0.4 | 73.0±0.4 | 71.5±0.6 | 74.2±0.4 |
| ConvNet | 100% | 64.0±2.3 | **91.8±0.2** | 89.2±0.8 | 84.4±0.6 | 88.8±0.8 | 91.1±0.4 |
| | 10% | 77.3±4.5 | 91.3±0.1 | **90.8±0.3** | 89.5±0.4 | 90.8±0.2 | 91.0±0.1 |
| | 5% | 77.1±8.2 | 91.4±0.3 | 90.6±0.9 | **90.0±0.2** | 90.8±0.1 | 91.3±0.1 |
| | 3% | **83.5±5.6** | 91.3±0.2 | 90.2±2.0 | 89.9±0.5 | **90.9±0.3** | **91.4±0.2** |
| LeNet | 100% | 71.8±3.4 | **80.1±0.4** | 76.6±3.8 | 77.8±1.3 | 78.5±1.6 | 76.5±1.1 |
| | 10% | **74.7±1.9** | 79.9±1.0 | 77.0±0.6 | **78.9±0.6** | **79.5±1.2** | 76.6±0.5 |
| | 5% | 72.8±2.7 | 78.9±1.4 | **77.3±0.8** | 77.4±1.1 | 78.8±0.6 | **76.8±1.4** |
| | 3% | 73.0±4.8 | 78.0±1.4 | 76.5±1.1 | 78.3±1.1 | 78.3±0.5 | 75.3±0.5 |
| AlexNet | 100% | 69.3±9.6 | **83.7±0.8** | 81.9±0.7 | 79.7±2.0 | 79.4±1.2 | 81.5±0.8 |
| | 10% | 72.7±9.4 | 82.8±0.3 | **82.0±0.3** | **82.6±0.3** | 82.4±1.0 | 81.4±0.4 |
| | 5% | 67.6±5.3 | 82.7±0.7 | 81.1±0.8 | 82.3±0.3 | 82.5±0.4 | **81.6±0.1** |
| | 3% | **74.4±8.9** | 82.8±0.2 | 80.6±2.2 | 82.1±0.5 | **83.0±0.2** | 81.2±0.4 |
| VGG | 100% | 71.7±3.2 | **89.4±0.3** | 89.0±0.4 | 88.6±0.4 | 88.7±0.5 | 89.1±0.4 |
| | 10% | 79.8±4.4 | 89.3±0.4 | **89.4±0.2** | **89.6±0.3** | **89.4±0.3** | 89.3±0.4 |
| | 5% | 79.3±6.0 | **89.4±0.2** | 89.3±0.4 | 89.3±0.2 | 89.3±0.2 | **89.4±0.1** |
| | 3% | **83.5±4.5** | 89.3±0.2 | 89.2±0.4 | 89.5±0.3 | 89.3±0.2 | 89.3±0.2 |
| ResNet | 100% | 81.0±1.8 | 89.0±0.2 | 88.9±0.2 | 88.4±0.3 | 88.2±0.7 | 88.5±0.2 |
| | 10% | 85.2±1.4 | **90.0±0.4** | **89.8±0.2** | **89.7±0.4** | 89.4±0.2 | **89.4±0.4** |
| | 5% | 84.1±3.4 | 89.5±0.1 | 89.4±0.3 | 89.1±0.3 | **89.7±0.1** | **89.4±0.3** |
| | 3% | **86.3±1.0** | 89.4±0.3 | 89.4±0.3 | 89.3±0.2 | 89.4±0.4 | 89.3±0.3 |

Table 18: Cross-architecture generalization on CIFAR10 and IPC=50 with DC algorithm.

| Sample Ratio | Evaluate Network | | | | | |
|---|---|---|---|---|---|---|
| | MLP | ConvNet | LeNet | AlexNet | VGG | ResNet |
| 100% (Full) | 28.01±0.40 | 54.02±0.51 | 28.12±2.16 | 29.48±0.58 | 39.44±0.67 | 22.72±1.08 |
| 70% | 29.48±0.28 | **55.96±0.40** | 30.83±1.51 | 29.54±2.60 | 41.99±0.47 | 24.35±0.42 |
| 50% | **30.40±0.28** | 55.25±0.32 | 31.15±1.25 | 30.53±2.21 | 43.09±0.41 | **25.81±1.04** |
| 30% | 30.15±0.21 | 54.77±0.47 | **31.44±0.72** | **33.45±1.18** | **43.35±0.60** | 25.48±0.53 |

# F  IMPLEMENTATION DETAILS

## F.1  DATASETS AND METRIC

Our experiments are conducted on the following datasets and we report the top-1 accuracy as the metric, most of which are widely adopted in dataset distillation.

- CIFAR10 (Krizhevsky et al., 2009): image dataset of common objects with 10 classes and 50,000 image samples. The images are 32x32 with 3 channels.

- CIFAR100 (Krizhevsky et al., 2009): image dataset of common objects with 100 classes and 50,000 samples. The images are 32x32 with 3 channels.

- MNIST (LeCun et al., 1998): hand-writing digit dataset with 10 classes and 60,000 samples. The images are 28x28 grayscale.

- SVHN (Netzer et al., 2011): street digit dataset with 10 classes and 73,257 samples. The images are 32x32 with 3 channels.

- TinyImageNet (Le & Yang, 2015): a subset of ImageNet with 200 classes and 100,000 images. The images are 64x64 with 3 channels.

- ImageNet (Deng et al., 2009): image datasets of common objects with 1000 classes and 1,281,167 samples. We resize the images to 64x64 with 3 channels following the previous setting (Zhou et al., 2022).

- Kinetics-400 (Carreira & Zisserman, 2017): human action video dataset with 400 classes and 215,617 video samples. The videos are resampled to 8 frames per clip and resized to 64x64.

Table 19: Cross-architecture generalization on CIFAR10 and IPC=10 with MTT algorithm.

| Sample Ratio | Evaluate Network | | | |
|---|---|---|---|---|
| | ConvNet | AlexNet | VGG | ResNet |
| 100% (Full) | 64.3±0.7 | 34.2±2.6 | 50.3±0.8 | 46.4±0.6 |
| 90% | **64.6±0.4** | **34.3±2.4** | **51.1±1.1** | **48.6±0.4** |

## F.2 NETWORK ARCHITECTURES

Following the previous work, in most of the experiments, we adopt ConvNetD3 as the network to probe the data. This network consists of 3 convolutional layers with a 3x3 filter, each of which has 128 channels and is followed by a ReLU non-linearity and an InstaceNorm layer. The average pooling layer aggregates the feature map to a 128d vector and produces the logit with a linear layer.

We also adopt other network architectures in Tab. 4 in the main paper, including MLP (three linear layers with hidden layer size 128), AlexNet (Krizhevsky et al., 2017), ResNet18 (He et al., 2016) (we use ResNet18+BatchNorm with average pooling for DC algorithm) and VGG11 (Simonyan & Zisserman, 2014) (we use VGG11+BatchNorm for DC algorithm).

## F.3 EXPERIMENTS OF RANDOM OR LOSS SELECTION

In Sec. 3.1 and Sec. 4.3 in the main paper, we extensively study the critical sample ratio by random or loss value. We mainly follow the default experiment settings given by each algorithm. The experiments are conducted on less than 4 TITAN Xp depending on the data size and IPC. We list the experiment details involving Tab. 1-8 in the paper:

1. For DC (Zhao et al., 2020) and DSA (Zhao & Bilen, 2021), on all datasets, we run the distillation for 1000 iterations with SGD optimizer and momentum 0.5. The number of inner loop and outer loop are (1, 1) for IPC=1, (10, 50) for IPC=10, (50, 10) for IPC=50. The learning rate of synthetic image and network are 0.1 and 0.01. The batch size for each class is 256 and when the sample ratio is low, we half the batch size until it is less than twice the largest class size. We use `color, crop, cutout, scale, rotate` DSA augmentation on all datasets and additional `flip` on the non-digit datasets. By default, `noise` initialization is used.

2. For DM (Zhao & Bilen, 2023), we run the distillation for 10000 iterations on TinyImageNet and 20000 iterations for the others with SGD optimizer and momentum 0.5. The learning rate of synthetic image and network are 1.0 and 0.01. The batch size for each class is 256 and when the sample ratio is low, we half the batch size until it is less than twice the largest class size. The same Siamese augmentation strategy is used as in the DSA experiments. By default, `real` initialization is used (the initial images are drawn after dropping).

3. For MTT (Cazenavette et al., 2022), we drop the same data samples for buffering and distillation. The expert trajectories are trained for 50 epochs for 100 repeats and we run the distillation for 10000 iterations. We appreciate and follow the detailed hyper-parameters provided by the authors.

4. For CAFE (Wang et al., 2022), as default, we run the distillation for 2000 iterations. The initial learning rate is 0.1 and decays by 0.5 at 1,200, 1,600, and 1,800 iterations. The weight of the inner layer matching loss is 0.01 and an additional loss weight of 0.1 is put on the matching loss of the third and fourth layers. `Noise` initialization is used.

5. For LinBa (Deng & Russakovsky, 2022), we run distillation for 5000 iterations with SGD optimizer with momentum 0.9. The inner steps of BPTT are 150 and the number of bases is 16. The learning rate of synthetic image and network is 0.1 and 0.01.

6. For IDC (Kim et al., 2022b), we use the "reproduce" setting of the opened source code, which automatically sets up the tuned hyper-parameters. We use multi-formation factor 2.

7. For KIP (Nguyen et al., 2020), we test on the finite-width model (KIP-NN) and use label learning. We use longer training steps for converged results.

8. For FRePo (Zhou et al., 2022), we use the official PyTorch implementation and the default parameters, except the learning rate of 0.001 and we run the distillation for 500,000 steps.

9. For HaBa (Liu et al., 2022), we follow the official instructions and adopt the parameters from MTT. And for the exclusive parameters for HaBa, we use the values given in the code.

10. For RFAD (Loo et al., 2022), we test on the finite-width model (ConvNet) and load the training hyperparameters for finite training results in the paper. The choice of label learning follows the remarks in the paper.

11. For IDM (Zhao et al., 2023), we thank the authors and we directly adopt the official running commands.

We try sample ratio of 0.02%, 0.04%, 0.1%, 0.3%, 0.5%, 1%, 3%, 5%, 8%, 10%, 15%, 20%, 30%, 40%, 50%, 60%, 70%, 80%, 90% to find the critical sample ratio. The removal of data samples is class-wise. Each experiment is repeated 5 times for mean $\mu$ and standard deviation $\sigma$ and we regard the experiment performance as comparable to the experiment on full data if its mean accuracy is within the $[\mu - \sigma, \mu + \sigma]$ of full data performance. Note that we do not present a finer-grained critical sample ratio since it requires too many computation resources (the bisection method is not applicable here) and our granularity is enough to demonstrate the data redundancy.

### F.4 COMPUTATION OF UTILITY INDICATORS

The parameters and settings of utility indicators are as follows:

**Classification Loss**    We train ConvNetD3 model on each dataset (ConvNetD3+GRU for Kinetics-400) for multiple trials for the loss indicator. We take the average loss curve of multiple trials. By default, we use a Gaussian filter with $\sigma = 3$ to smooth the loss curve and take the loss value at the last epoch, which is approximately equivalent to the **weighted mean loss value of the last 8 epochs**. The training details are:

- CIFAR10: 50 trials for 100 epochs with learning rate 3.0e-3 and batch size 512.
- CIFAR100: 50 trials for 250 epochs with learning rate 5.0e-3 and batch size 512.
- MNIST: 50 trials for 50 epochs with learning rate 3.0e-4 and batch size 512.
- SVHN: 50 trials for 100 epochs with learning rate 1.0e-3 and batch size 512.
- TinyImageNet: 50 trials for 100 epochs with learning rate 5.0e-3 and batch size 512.
- ImageNet: 30 trials for 20 epochs with learning rate 3.0e-3 and batch size 256 (early stop).
- Kinetics-400: 10 trials for 20 epochs with learning rate 1.0e-2 and batch size 128 (early stop).

Note that considering the conclusion in Sec. 4.3, we adopt an early stop on large-scale datasets to reduce the training cost.

**Data Density**    We use the same parameters for Classification Loss to train the classifier and extract the GAP (global average pooling) feature for each sample. Within each class, we use the mean distance of each sample to its K-nearest neighbors to represent the data density. We use K=100 for CIFAR10, MNIST, SVHN, and K=20 for CIFAR100 and TinyImageNet due to their smaller IPC.

**Distance to the Cluster Center**    Following Liu et al. (2023), we cluster each class with K-Means into K clusters, and for each sample, we compute its distance to the nearest cluster center. We use K=128 for CIFAR10, MNIST, SVHN (same as Liu et al. (2023)), and K=16 for CIFAR100 and TinyImageNet due to their smaller IPC.

**Distance to the Initial Image**    For each sample, we compute its distance to the nearest initialization image in the same class. Therefore, we use `real` initialization for all the involved experiments.

**Monte-Carlo Indicator**   We use the DC algorithm and IPC=1 to probe the data utility for the Monte-Carlo indicator. The obtained utility value also shows generalizability to different algorithms or IPCs (see Tab. 12) The experiment details are as follows:

- CIFAR10: sample ratio $M/N = 0.05$, repeated for 1,000 trials.

- CIFAR100: sample ratio $M/N = 0.3$, repeated for 500 trials.

- MNIST: sample ratio $M/N = 0.008$, repeated for 1,000 trials.

- SVHN: sample ratio $M/N = 0.3$, repeated for 500 trials.

Though the number of trials is still insufficient for the convergence of indicators, it still takes 50-100 GPU hours, showing its inefficiency.

### F.5   EXPERIMENTS OF THE PROPOSED BASELINE

We propose a simple baseline in Sec. 5.1 in the main paper. For both algorithms, we prune the data batch after 40% of the outer loops for a more accurate loss indicator. For the DC algorithm, we use the same hyper-parameters as the previous experiments, except that when IPC=1, we run 100 iterations and 10 outer loops for each iteration (the default settings at IPC=1 do not update the network). For the IDC algorithm, we use multi-formation factor 3 and we run the distillation for 500 iterations with 3x of the default learning rate. By default, our pruning rate is 30%, while we use 10% for DC algorithm on CIFAR10+IPC=10, CIFAR100+IPC=1, SVHN+IPC=10, use 10% for IDC on CIFAR100+IPC=10, SVHN+IPC=10, and use 15% for IDC on CIFAR10+IPC=1/10.

### F.6   EXPERIMENTS ON THE LARGE-SCALE DATASETS

We apply our selection paradigm on larger-scale datasets in Sec. 5.2 in the main paper. The experiments are conducted on at most 4 RTX 3090 GPUs and the details are as follows:

- ImageNet, DC: the training of DC exceeds the usual GPU capacity so in compromise we separate the 1000 classes into ten 100 class splits, which will slightly decrease the accuracy. The other hyper-parameters are the same as the previous experiments. For our paradigm, we prune 50% samples and early stop at 800 iterations due to its faster convergence.

- ImageNet, DM: the DM algorithm is safe for class-separate training so we separate the classes into 4 splits at IPC=1, 8 splits at IPC=10, and 20 splits at IPC=50. We run the distillation for 5000 iterations with a learning rate of 5.0. For our paradigm, we prune 50% samples and early stop at 2,000, 3,000, and 3,000 iterations for IPC=1/10/50 respectively.

- ImageNet, MTT: the expert trajectory is too expensive to compute so we only run MTT with our selection method. We prune 90% samples which reduces 84% of the trajectory training time. We train 60 trajectories for 50 epochs. MTT also requires large GPU memory due to the unrolling of back-propagation, so we use synthetic steps=5, expert epochs=2, and maximum start epoch=5. We run the distillation for 5000 iterations with an image learning rate of 30,000 and a step size learning rate of 1.0e-6.

- Kinetics-400, DM: on Kinetics, we run DM for 5000 iterations with a learning rate of 5.0 and batch size of 128. We separate the classes into 8 splits at IPC=1 and 20 splits at IPC=10. For our paradigm, we prune 50% samples and early stop at 4000 iterations. We do not use DSA augmentation for Kinetics.

- Kinetics-400, MTT: we prune 90% samples and train 40 trajectories for 50 epochs with batch size 128. We use synthetic steps=5, expert epochs=2, and maximum start epoch=5. We run the distillation for 5000 iterations with an image learning rate of 30,000, step size learning rate of 1.0e-6, real batch size 128, and synthetic batch size 64. We do not use DSA augmentation for Kinetics.

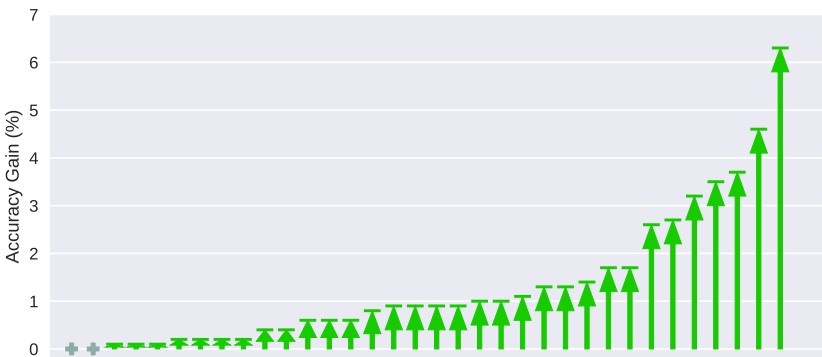

Figure 9: Performance gain of data dropping by loss values. Each arrow is an experiment trial in Tab. 7 in the main paper, and its length indicates the distillation accuracy gained after dropping part of the samples.

# G    MORE VISUALIZATIONS

## G.1    PERFORMANCE GAIN OF DATA DROPPING BY LOSS

We demonstrate and discuss the performance gain brought by data dropping in Tab. 7 in the main paper, where dropping by loss could consistently and significantly enhance the distillation performance. To make it more intuitive, we plot the performance gain in Fig. 9, which is notable.

## G.2    EXAMPLES OF SAMPLES AT DIFFERENT UTILITY LEVELS

In this section, we present some data samples at different utility levels to qualitatively compare the utility indicators and the datasets.

### G.2.1    COMPARISON AMONG UTILITY INDICATORS

To compare the utility indicators, we stratify the CIFAR10 dataset into 10 layers according to various indicators and visualize some samples in the layer with the smallest or largest utility in Fig. 10. Note that the initialization indicator is not considered since it depends on the specific initialization value of the distillation algorithm. As shown in the figure, the samples selected by loss, density, and clustering indicators show some similarities. The samples with small utility are noisy and usually hard and corner cases, *e.g.*, only part of the birds are shown, or some dogs are acting in strange poses, or the images of ships are captured with unusual viewing angles. Meanwhile, the samples with large utility are easy cases that have ideal saliency, viewing angle, and clean background. However, the selection principle of the Monte-Carlo indicator is not as clear as the rest three. We speculate that the Monte-Carlo indicator has partially taken the data diversity into account and thus the subset with large utility is more diversified.

### G.2.2    COMPARISON AMONG DATASETS AND DATA DIVERSITY

We also show data samples of various datasets at different utility levels. Similarly, we stratify the datasets into 10 layers according to the loss indicators and visualize some samples in the layer with the smallest or largest utility in Fig. 11. A similar conclusion can be drawn that the samples with small utility are noisy and those with large utility are easier cases, including the large-scale datasets (ImageNet and Kinetics-400). The digit datasets (MNIST, SVHN) show significantly more *diversity vanishing* than the rest realistic datasets. Moreover, the diversity vanishing issue is mild for large-scale datasets such as ImageNet since the intra-class discrepancy is large such that any subset is diversified enough.

To extend our discussion on the data diversity (Sec. 4.4 in the main paper), we give some more examples to compare the diversity for data strata with different loss values in Fig. 12 on MNIST.

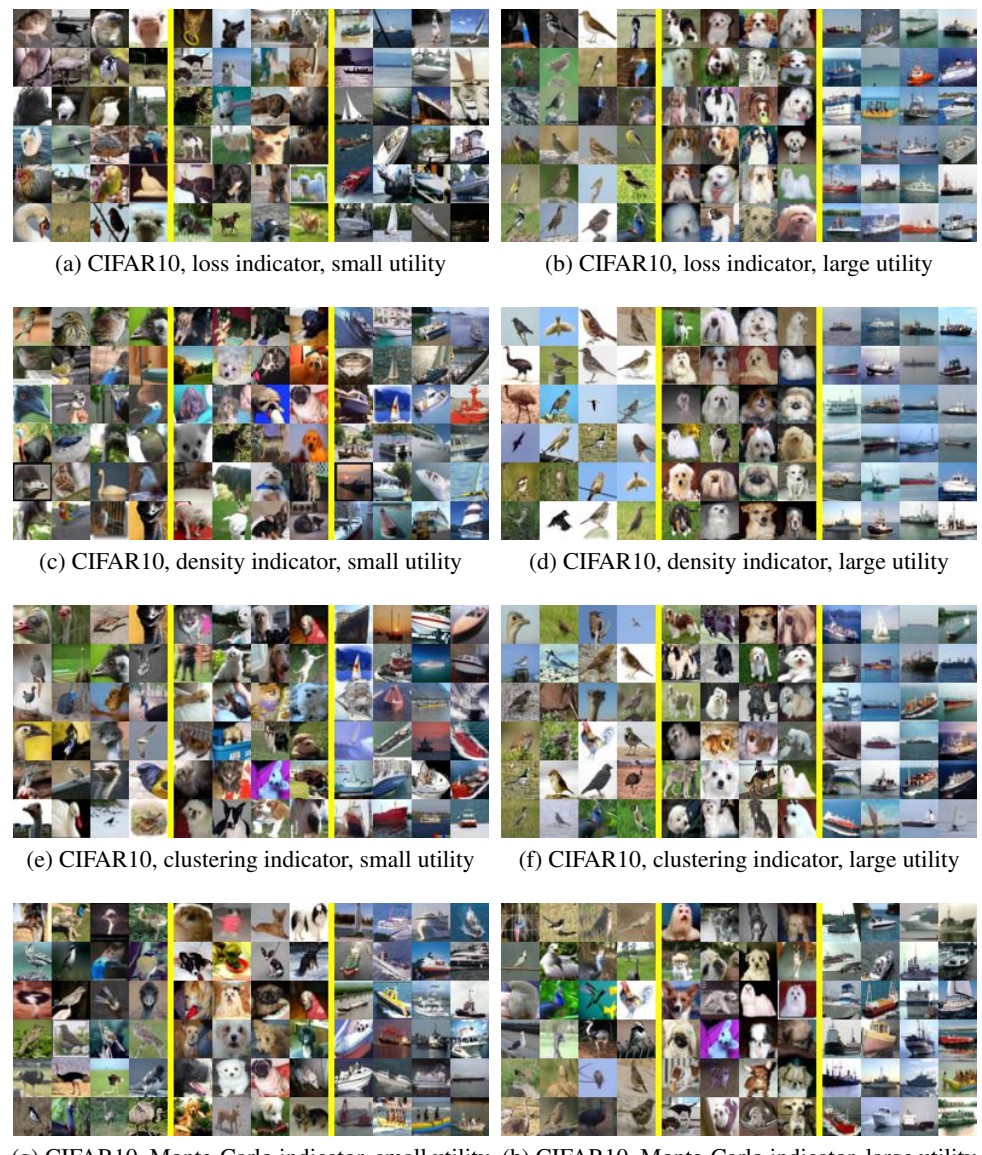

(a) CIFAR10, loss indicator, small utility  (b) CIFAR10, loss indicator, large utility

(c) CIFAR10, density indicator, small utility  (d) CIFAR10, density indicator, large utility

(e) CIFAR10, clustering indicator, small utility  (f) CIFAR10, clustering indicator, large utility

(g) CIFAR10, Monte-Carlo indicator, small utility  (h) CIFAR10, Monte-Carlo indicator, large utility

Figure 10: Qualitative comparison of utility indicators. We stratify CIFAR10 and show samples in the layers with the smallest utility (left column) or largest utility (right column). We show 3 classes for each layer.

The groups with large loss values are mainly corner cases. Furthermore, as the loss value decreases (S7 or S10), the diversity significantly drops as shown in Fig. 12 (c, d, g, h).

## H  LIMITATIONS

In this work, we thoroughly study and discuss the data utility. To analyze the data redundancy clearly, we focus on a simplified scenario to make it more applicable, thus some limitations still exist.

**Higher-order interaction of data utility.** In this work, we mainly discuss the data utility from a mean-field view, so the interactions of data utility between samples are omitted. A detailed discussion of high-order interaction in data utility has been given in Sec. 4.4 in the main paper.

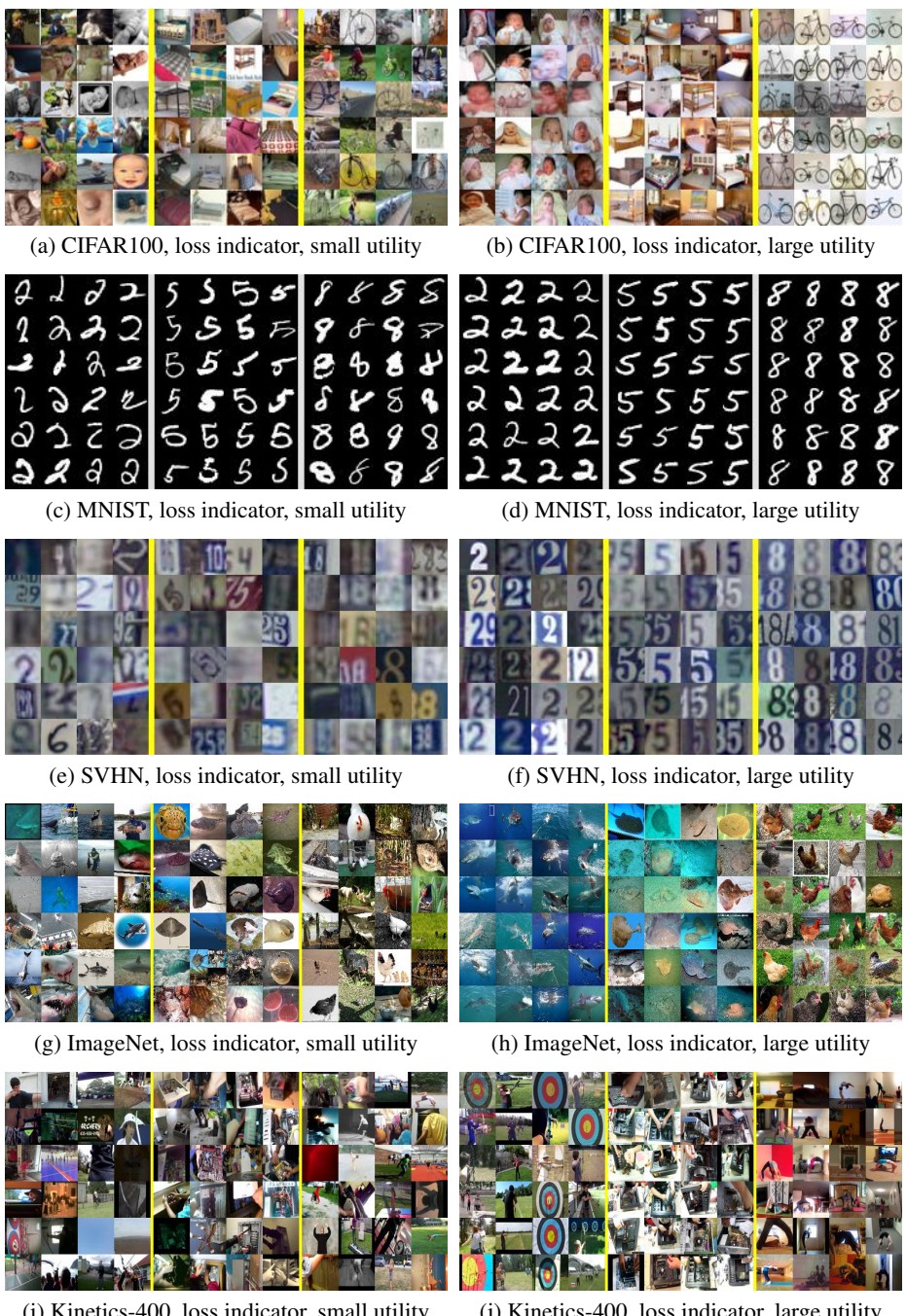

(a) CIFAR100, loss indicator, small utility      (b) CIFAR100, loss indicator, large utility

(c) MNIST, loss indicator, small utility      (d) MNIST, loss indicator, large utility

(e) SVHN, loss indicator, small utility      (f) SVHN, loss indicator, large utility

(g) ImageNet, loss indicator, small utility      (h) ImageNet, loss indicator, large utility

(i) Kinetics-400, loss indicator, small utility      (j) Kinetics-400, loss indicator, large utility

Figure 11: Qualitative comparison of multiple datasets. We conduct stratified experiments with loss indicators and show samples in the layers with the smallest utility (left column) or largest utility (right column). We show 3 classes for each dataset.

**Variable data utility during training.** In our study, we assume the data utility is an overall indicator of the importance of samples, therefore it is a constant throughout the entire training process. However, in real scenarios, the data utility may vary during training, *i.e.*, it is a function of training time $U(\mathcal{X}, t)$. *e.g.*, the data utility of some easy cases may diminish during training since they are easy for the model to fit. However, considering training time will significantly complicate the model and we leave it to future work.

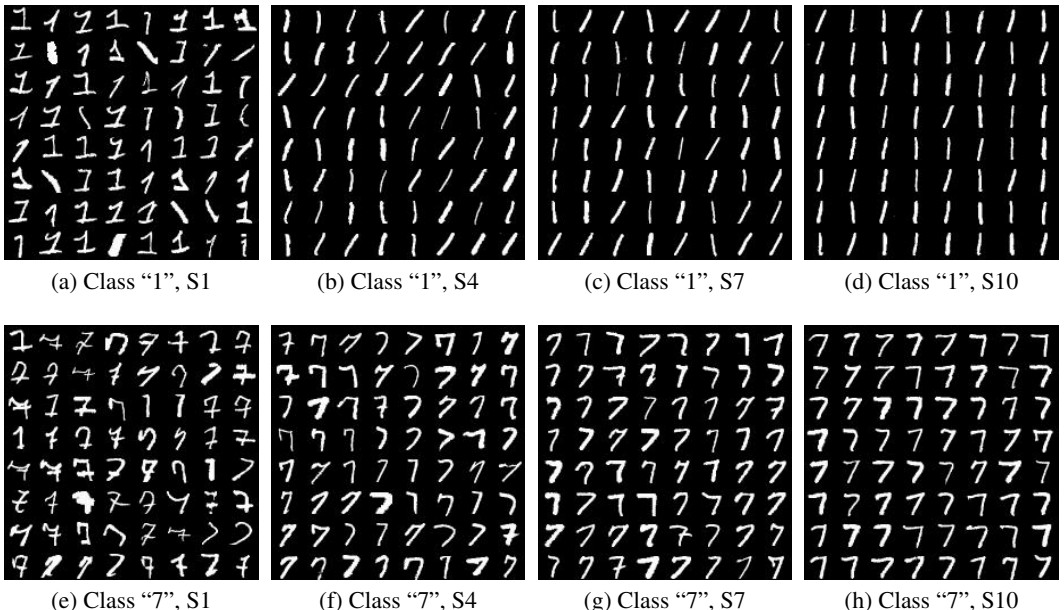

(a) Class "1", S1    (b) Class "1", S4    (c) Class "1", S7    (d) Class "1", S10

(e) Class "7", S1    (f) Class "7", S4    (g) Class "7", S7    (h) Class "7", S10

Figure 12: More examples of different strata in the MNIST dataset. The data are stratified by classification loss. The samples in S1 have the lowest loss values and those in S10 have the largest loss. The diversity significantly drops when the sample loss decreases (*e.g.*, S7, S10).

## I    LICENSES

Here are the source and license of the assets involved in our work. We sincerely appreciate and thank the authors and creators.

**Datasets:**

- CIFAR10, CIFAR100 (Krizhevsky et al., 2009): URL, unknown license.
- MNIST (LeCun et al., 1998): URL, MIT License.
- SVHN (Netzer et al., 2011): URL, unknown license.
- Tiny-ImageNet (Le & Yang, 2015): URL, unknown license.
- ImageNet (Deng et al., 2009): URL, custom license, research, non-commercial.
- Kinetics-400 (Carreira & Zisserman, 2017): URL, Creative Commons Attribution 4.0 International License.

**Code:**

- DC (Zhao et al., 2020), DSA (Zhao & Bilen, 2021), DM (Zhao & Bilen, 2023): URL, MIT License.
- MTT (Cazenavette et al., 2022): URL, MIT License.
- CAFE (Wang et al., 2022): URL, no license yet.
- IDC (Kim et al., 2022b): URL, MIT License.
- LinBa (Deng & Russakovsky, 2022): URL, no license.
- CORDS: URL, MIT license

