# OpenReview forum: "Distill Gold from Massive Ores: Efficient Dataset Distillation via Critical Samples Selection"
_ICLR.cc/2024/Conference — ICLR 2024 Conference Withdrawn Submission_

### Official Review · Reviewer_yzKj · 2023-10-29

**Soundness:** 3 good
**Presentation:** 3 good
**Contribution:** 3 good
**Rating:** 5
**Confidence:** 4

**Summary:**

The paper proposes a data pruning algorithm for efficient dataset distillation. Multiple indicators are proposed to find those important samples for dataset distillation. Experiments are conducted on several baseline methods, and the results show that at least 10% of the dataset can be discarded while not harming the performance.

**Strengths:**

1. The presentation and the writing quality of the paper is good.
2. The amount of experiments is rich and sufficient.
3. The topic is important and the authors have extended dataset distillation to video data which are not done by others before.

**Weaknesses:**

1. I think the loss indicator is also very heavy in computation because according to the description it requires 50~100 epochs of training for 50 trails. The Monte-Carlo indicator also seems to require a long running time. It would be greatly appreciated if the authors can provide running time comparison. While Section 4.3 provides some ways to mitigate this, but I cannot find what would be the final performance using this "economic" version of algorithm.

2. A red flag for me is that, it seems that the proposed methods become less effectiveness when the dataset distillation methods are becoming stronger. More specifically, MTT has higher performance compard to DC and DSA, but it also requires more data to achieve the same level of performance. There could be a chance that this method will become less effective, if not effective at all. I would suggest the authors to benchmark on more advanced methods such as RFAD [r1]  to show the proposed method can still be effective. For example, why did the authors not report performance on IDC in Table 7, while providing IDC's performance in somewhere else like Table 6.

3. How to decide the data dropping ratio in principle? Deciding such a threshold could also bring overhead.

4. Table 9 shows a simple twist can help improve the performance but it seems that the gaps will be drastically closed when the number of IPC increases.

5. It would be better if the authors can provide results of IPC=10 / 50 in Table 15 to show that the dataset can be pruned at higher budgets.

[r1] Efficient Dataset Distillation using Random Feature Approximation

**Questions:**

Please refer to above section.

---

### Official Review · Reviewer_dAWG · 2023-10-30

**Soundness:** 2 fair
**Presentation:** 1 poor
**Contribution:** 2 fair
**Rating:** 3
**Confidence:** 3

**Summary:**

This paper is on dataset distillation: reducing a dataset down to much fewer examples, such that training on this smaller set (which may be synthetic) results in a model of similar performance as the original. The authors introduce a meta layer to the problem, investigating how using a smaller proportion of the original dataset can accelerate the process of distillation. In particular, they observe that much fewer examples are needed for distillation, proposing several measures “data utility” that allow for selecting which sample to keep. Experiments with a number of dataset distillation methods (using less samples) on several benchmark datasets (including a couple large scale ones: ImageNet-1K, Kinetics-400) illustrate that indeed the proposed methods of selecting samples can effectively reduce the number of samples used, in some case outperforming using the whole dataset.

**Strengths:**

## S1. Dataset size reduction
This paper proposes reducing the amount of data used for dataset distillation to accelerate these methods. In many cases, this is pretty significant; the sound bite the authors use is 0.04% of the full dataset, which while not representative of all scenarios, is nonetheless impressive. This makes them more usable and saves computation. As the authors point out, some of the recent methods are significantly slower than just training the model on the whole dataset. This paper may serve as a good reference for dataset distillation practitioners for doing so more efficiently. See however W2c.

## S2. Experiments
The authors demonstrate their findings primarily on 4 dataset distillation methods (with some additional experiments on 3 others) and a number of datasets. Most impressively, they consider datasets that are of relatively large scale (ImageNet-1K + Kinetics-400). Given how large these datasets (I believe ImageNet is 150 GB), the thought of being able to distill these down to something quite small is quite compelling. However, see W3a.

## Miscellaneous:
- I do like that the authors go beyond the mean-field assumption and look at higher-order interactions, like the impact of diversity. In the extreme case, a high utility sample that is duplicated in the dataset probably shouldn’t be included twice. I am a little disappointed though that the authors more or less punted on a more thorough investigation for future work.

**Weaknesses:**

## W1. Technical rigor
While there are attempts to provide a formal treatment to the problem, the technical rigor of this paper is not satisfactory. \
a) The terms “utility” and “capacity” are commonly utilized throughout the paper to describe the value of a particular sample for distillation and how much information a synthetic distilled dataset can hold, including as part of formal definitions. However, the precise definitions of these terms are in fact quite vague and unclear, instead resembling pseudo-information theory. This paper’s usage of utility and capacity in definitions and equations is not appropriate. \
b) Concepts of “critical sample size” and “critical sample ratio” are introduced, by these are also not precisely defined, which renders the results in Table 1-4 seemingly arbitrary. \
c) There are a number of other issues with notation and some of the tables/figures. See Miscellaneous + Questions below.

## W2. Novelty + Significance
a) The proposed method more or less boils down to doing dataset distillation on a subset of the data. The fact that this is more efficient than doing it on the whole dataset isn’t particularly surprising, and that careful selection of data is better than random selection is in line with existing works on data pruning (e.g. [a]). \
b) Loss as a measure of utility is more or less the same as techniques commonly used in active learning, or several branches of approaches continual learning (e.g. coreset selection in replay-based approaches, or importance weighting in regularization-based ones [b]). These similarities aren't discussed in the current draft, but should be. \
c) The second paragraph of the Introduction is a written a little too harshly. While it’s true that data distillation taking a long time is a bad thing, that doesn’t render it completely unusable as a technique, as there are still applications where the portability of data matters more than having a trained model. Some may argue that the primary utility of dataset distillation is data portability; in many cases, distillation can be done offline, so how long it takes, while still important, is less of a concern. Also, while the “100x longer” statistic is eye-catching, this is also a cherry-picked example that is especially bad, and not necessarily representative of dataset distillation as a whole.

## W3. Experiments
a) One of the headliner claims of this paper is that it is able to extend to larger-scale datasets like ImageNet-1K and Kinetics-400. However, the ImageNet accuracy is ~2% for 1 IPC, and ~9% for 50 IPC. While I understand that this is compressed to a much smaller synthetic set and better than the baseline, this is not anywhere close to how modern methods perform on ImageNet, and not what I consider as “working”; same goes for Kinetics-400. Simply running a method on a dataset regardless of results is not the same as saying the method works well on it. As such, the claims here are misleading. \
b) A common use case of dataset distillation is continual learning. It could have been nice to see some experiments in this setting.

## W4. Writing
The writing could use some improvement. There are a number of grammatical or idiomatic errors, and in a number of instances, the word choice implies the wrong thing, which can be distracting. I list out some examples below under “Miscellaneous”, but this is not an exhaustive list. I recommend the authors give this manuscript another careful round of edits.

The organization of this paper is also somewhat non-standard. Experiments are mixed in through Section 3 (Preliminaries) and 4 (Estimating Data Utility). There is no independent Experiments section. This isn’t necessarily a requirement for writing a paper, but following a more typical separation will help readers more easily find particular content. For example, the current organization is almost chronological in nature, adding new wrinkles or changes to the methodology after pervious benchmarking had already been done. This makes it hard to tell what finally this paper’s contributions are.

## Miscellaneous:
- pg 1: “has become critical which enables high performance” <= awkward wording
- pg 1: “reduce the model size” <= extraneous “the”
- pg 1: “maintaining the model performance trained on the synthetics” <= I think this is actually trying to say that the goal is to be able train a model on the small synthetic dataset that has similar performance to training the real large dataset.
- pg 1: “data utilization” doesn’t seem to be the heart of the issue the authors are seeking to address. If I understand correctly, the authors are more concerned about the efficiency of data compression
- pg 1: “ a natural assumption comes up” <= assumptions are made by choice. Also, what follows is not an assumption, but rather a hypothesis/design choice.
- Figure 1 right: Rather than have points off the chart, can this be shown in log scale instead?
- pg 3: “we first argue that data redundancy is extremely severe in distillation” <= This is not a property of distillation, but rather the dataset for the specific learning task.
- pg 3: “to support the general the observation”
- pg 4: Tables 1-4: Methods/architectures should be cited.
- pg 4: “… real dataset and we have more freedom …” <= run-on sentence
- pg 4: “… high computation cost and MTT exceeds …” <= run-on sentence
- Figure 3 is confusing. Why does the subset have the same utility as the full dataset? Why does the model have more utility than the synthetic bottleneck?
- Definition 3: Cardinality is more commonly expressed with vertical bars like $|\mathcal X|$.
- pg 4: “A Larger” <= second capitalized word
- pg 6: Using $M$ for the size of the subset and $m$ for the size of the whole dataset feels backwards.

[a] Sorscher, Ben, et al. "Beyond neural scaling laws: beating power law scaling via data pruning."  NeurIPS 2022. \
[b] Kirkpatrick, James, et al. "Overcoming catastrophic forgetting in neural networks." PNAS 2017.

**Questions:**

Q1: pg 2: What is CIFAR10 for DC? Acronym isn’t defined. What does instance-per-class mean? \
Q2: Definition 1: Shouldn’t the critical sample size be a function of $\epsilon$? What is the meaning of “comparable distillation accuracy” in Table 1. I imagine the critical sample size/ratio can vary dramatically depending on how we define “closeness”. \
Q3: Tables 1 + 2: There are several instances where the minimal data ratio for higher IPC is lower. Why does this happen, and why for these specific examples (e.g. CAFÉ for CIFAR10 + MNIST, or MTT for CIFAR 10)? Also why do so many of these values happen to have percentages that are multiples of 10? \
Q4: What is $N$? \
Q5: Fig 4: What is the x-axis of this figure? What is each arrow? \
Q6: After finding the data samples with the highest utility, how much value does dataset distillation provide? Can you just directly do few-shot learning on these samples? This is an important baseline to compare against.

---

### Official Review · Reviewer_DpjX · 2023-10-31

**Soundness:** 3 good
**Presentation:** 3 good
**Contribution:** 3 good
**Rating:** 6
**Confidence:** 4

**Summary:**

The paper tackles the computational inefficiency in dataset distillation, aiming to create smaller but informative datasets for machine learning models. The authors introduce a strategy to select high-utility training samples, reducing training costs without sacrificing performance. They propose methods for utility estimation and validate their approach through experiments on large and diverse datasets like ImageNet-1K and Kinetics-400. The paper claims that this technique not only speeds up distillation but can also sometimes improve model performance.

**Strengths:**

1. **Originality**: The paper tackles the problem of dataset distillation across a range of data types, including videos, which is relatively uncommon in this domain.
2. **Quality**: The experimental setup is robust, with tests conducted on diverse datasets, neural networks, distillation algorithms, initialization methods, and synthetic data sizes (IPC). This exhaustive approach adds credibility to the paper's findings.

**Weaknesses:**

1. **Incomplete Results**: Some of the tables lack IPC-50 results for specific datasets, creating an inconsistency in the presentation of the findings.
2. **Clarity on Data Selection Methods**: It would be beneficial to clarify how the data selection methods were compared. For instance, it is unclear if all samples were selected at the beginning of the training or if the selection was epoch-based.

**Questions:**

1. **Clarification on Cost Metrics:** The introduction mentions a 100x cost for MTT. Could you please elaborate on how this cost is calculated? Is it computational cost, time, or something else?
2. **Training Directly on Small Sample Size:** You mention that the method is capable of dataset distillation down to very small sets. What would be the performance if one were to train a model directly on these small sets without distillation?
3. **Data Selection Methodology:** Could you clarify how the comparison to data selection methods was conducted? Specifically, were all the distilled samples selected at the beginning of training, or was the selection made dynamically at each epoch?
4. **Coreset Selection Discussion:** The paper briefly touches upon coreset selection methods for comparison but lacks a substantive discussion or citation of relevant works in this area. For a more comprehensive treatment, you may consider referencing and discussing more recent publications on coresets, such as [1*] and [2*], as well as works that integrate coreset selection with dataset condensation like [3*].
5. **Utility Function Definition:** The concept of the utility of a dataset ($U(D)$) is introduced but not adequately defined. Could you provide a more formal definition? Is it task-dependent, perhaps reflected by the average test accuracy?
6. **Inconsistent Reporting of IPC-50 Results:** IPC-50 results appear for CIFAR-10 in Tables 1, 5, and 7, but they are missing for CIFAR-100, SVHN, and TinyImageNet in Tables 2, 6, and 9. Could you please provide these missing results for a comprehensive comparison?

[1*] Pooladzandi, O., Davini, D. & Mirzasoleiman, B.. (2022). Adaptive Second Order Coresets for Data-efficient Machine Learning. *Proceedings of the 39th International Conference on Machine Learning*, in *Proceedings of Machine Learning Research* 162:17848-17869 Available from https://proceedings.mlr.press/v162/pooladzandi22a.html.

[2*] Yang, Y., Kang, H. & Mirzasoleiman, B.. (2023). Towards Sustainable Learning: Coresets for Data-efficient Deep Learning. *Proceedings of the 40th International Conference on Machine Learning*, in *Proceedings of Machine Learning Research* 202:39314-39330 Available from https://proceedings.mlr.press/v202/yang23g.html.

[3*] Shin, S., Bae, H., Shin, D., Joo, W. & Moon, I.. (2023). Loss-Curvature Matching for Dataset Selection and Condensation. *Proceedings of The 26th International Conference on Artificial Intelligence and Statistics*, in *Proceedings of Machine Learning Research* 206:8606-8628 Available from https://proceedings.mlr.press/v206/shin23a.html.

---

### Official Review · Reviewer_ne7k · 2023-11-06

**Soundness:** 3 good
**Presentation:** 3 good
**Contribution:** 2 fair
**Rating:** 5
**Confidence:** 4

**Summary:**

The paper proposes a data-efficient dataset distillation method based on the concept of data utility, which measures the value or quality of data samples for distillation, and introduces various indicators to estimate the data utility and proposes a greedy selection strategy to find the optimal subset of real data. This simple plug-and-play mechanism is able to exploit the data utility during runtime, also can significantly reduce the training cost and enable distillation on large-scale and heterogeneous datasets.

**Strengths:**

- The paper provides a novel perspective on dataset distillation inspired by information theory and addresses the problem of data redundancy in distillation.
- The paper conducts extensive experiments and comparisons to validate the effectiveness and efficiency of the proposed data utility indicators and selection methods.
- The paper demonstrates that the proposed method can enhance the performance of existing distillation algorithms and extend them to more challenging scenarios such as ImageNet-1K and Kinetics-400.

**Weaknesses:**

- Hope to see a clear theoretical analysis or justification for the proposed data utility indicators and selection methods. It is not clear how they relate to the information content or transferability of data samples.

- The paper does not show significant performance improvement over the state-of-the-art distillation algorithms, please check the sota methods in: https://github.com/Guang000/Awesome-Dataset-Distillation . The performance gains are mostly marginal.

**Questions:**

- How does the proposed method compare with the DREAM and what are the main differences and advantages of the proposed method over DREAM?
- How does the proposed method handle the trade-off between data utility and data diversity? Is there a risk of overfitting to a small subset of real data that may not capture the full complexity and variability of the original dataset?
- How does the proposed method deal with the potential bias or noise introduced by the loss value as a utility indicator?